# Skeletons in the permafrost: Exploring climate-driven heritage loss and occupational health at the early modern whaling burial site of Likneset, Svalbard

Lise Loktu[1]*, Elin Therese Brødholt[2]

**1** High North Department/FRAM – High North Research Centre for Climate and the Environment, Norwegian Institute for Cultural Heritage Research (NIKU), Tromsø, Norway, **2** Department of Forensic Sciences, Oslo University Hospital, Oslo, Norway

* lise.loktu@niku.no

## Abstract

Rapid Arctic warming is accelerating the degradation of permafrost-preserved archaeological sites, threatening both cultural heritage and the scientific information it contains. Early modern whaling burial sites on Svalbard are particularly vulnerable due to their organic-rich burial contexts and exposed coastal settings. This study presents an integrated taphonomic and osteological analysis of the Likneset whaling burial site in northwestern Svalbard, addressing climate-driven changes in preservation conditions (RQ1) and the embodied health costs of labour in early modern Arctic whaling (RQ2). The material comprises 20 individuals excavated across three phases (Phase I: 1985–1990, n = 14; Phase II: 2016, n = 3; Phase III: 2019, n = 3). The dataset allows comparison across excavation phases and between contrasting geomorphological settings within the burial site, distinguishing erosion-exposed areas (A) from a more stable central zone (B). Burial structures, coffins, skeletal remains, and textiles were assessed using a unified preservation scoring system, combined with osteological analyses of demography, metabolic disease, developmental stress, musculoskeletal degeneration, and trauma. Results show that preservation at Likneset is structured by local geomorphology and ongoing environmental change. Burials in erosion-exposed areas display extensive disturbance and loss of organic materials, while graves in more stable settings retain better-preserved structures, skeletons, and textiles. Comparison of closely spaced burials within the same erosion-exposed area, excavated several decades apart, documents continued in situ degradation, with textile materials declining more rapidly than skeletal remains. Osteological evidence indicates a largely homogeneous burial population composed predominantly of young adult men. Despite generally robust stature, the skeletal record documents widespread physiological stress, including metabolic disease, developmental stress markers, and extensive activity-related skeletal changes developing

**Data availability statement:** All relevant data are within the paper and its Supporting Information files.

**Funding:** Lise Loktu received funding (NOK 550,000; grant 23/33) from the Svalbard Environmental Protection Fund to conduct new osteological analyses of skeletons from the whaling period (17th–18th century), excavated at the Likneset burial ground (ID 93705) in Smeerenburgfjorden during the 1985–1990 field campaigns (https://www.miljovernfondet. no/en/front-page/).

**Competing interests:** The authors have declared that no competing interests exist.

early in adulthood. Trauma is predominantly healed, suggesting that mortality was more closely linked to cumulative nutritional deficiency and prolonged physical strain. The results highlight growing challenges for heritage management on Svalbard, where strategies based on in situ preservation and managed decay are increasingly strained under warming permafrost conditions, underscoring the need for systematic monitoring, targeted documentation, and integration of archaeological data into climate adaptation planning before irreplaceable archives are lost.

## Introduction

### Climate change and archaeological preservation in the Arctic

Rapid climate warming is transforming Arctic environments at a rate far exceeding the global average [1], with profound consequences for archaeological heritage preserved in permafrost [2, 3]. Rising temperatures, permafrost thaw, coastal erosion, and increased moisture availability destabilise archaeological contexts [4–6] and accelerate the decay of organic materials such as wood, textiles, and human remains [7–15]. As a result, Arctic archaeological sites, long regarded as exceptionally well preserved, are increasingly subject to rapid and irreversible degradation, threatening both cultural heritage and the scientific information embedded within it [2,5].

On the Svalbard archipelago, located in the High Arctic between mainland Norway and the North Pole (Fig 1), these processes are particularly pronounced. Svalbard is one of the fastest-warming regions globally [16, 17], with documented thickening of the active layer and intensified coastal erosion over recent decades [18–20]. Much of the archipelago's cultural heritage is situated in low-lying permafrost terrain along fjords and coastlines, where thaw-related ground instability and marine erosion intersect and reinforce each other [19,21–24]. Increased visitor pressure further contributes to site disturbance at many fragile heritage sites [25–32].

Despite growing recognition of this vulnerability, archaeological sites remain weakly integrated into climate monitoring and adaptation frameworks on Svalbard, and long-term datasets documenting preservation change are largely absent [33–35]. This limits effective heritage management and our understanding of the past, and reduces the potential of archaeological archives to contribute empirical, time-depth perspectives on environmental change in Svalbard. In this context, sites that have been excavated repeatedly over time provide rare opportunities to document preservation trajectories directly within the archaeological record.

### Early modern whaling burial sites as climate-sensitive archives

Early modern whaling burial sites on Svalbard represent one of the most vulnerable categories of Arctic archaeological heritage [34,36]. Dating primarily to the seventeenth and eighteenth centuries, these burial sites are associated with some of the earliest large-scale European extractive activities in the High Arctic and are typically located in exposed coastal settings [37–42].

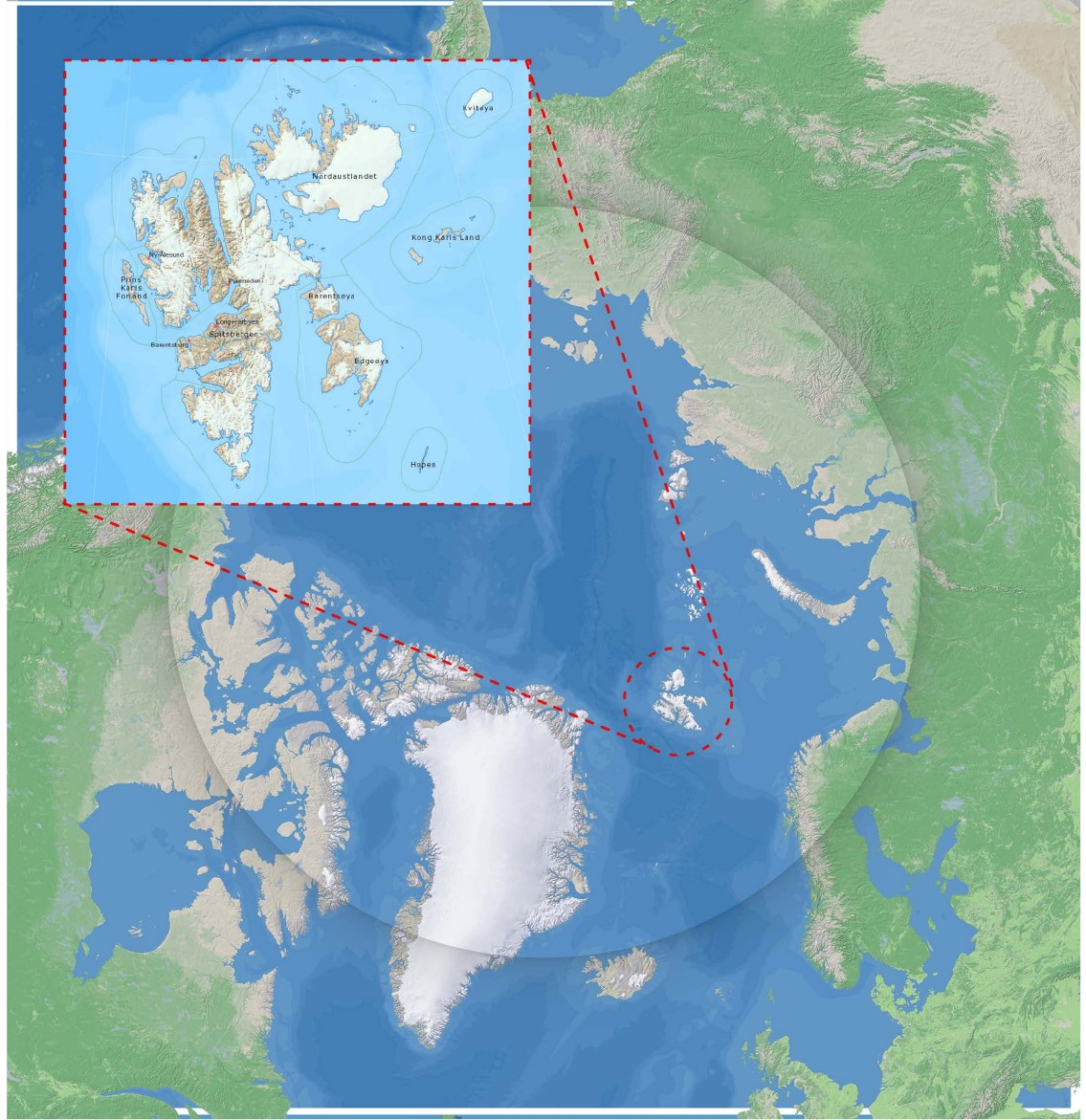

**Fig 1. Location of the Svalbard archipelago in the High Arctic.** Regional map showing Svalbard's position within the circumpolar north and its broader Arctic setting. Figure by Lise Loktu, NIKU. Basemap: Shaded Relief, public domain by Tom Patterson, and TopoSvalbard, Norwegian Polar Institute.

Preserved in permafrost, excavations conducted in the Smeerenburgfjorden region of northwestern Svalbard (Fig 2) during the late twentieth century revealed exceptionally well-preserved skeletal remains, textiles, and burial contexts. Together, these materials constitute a globally rare archaeological assemblage, offering unique insights into the lives, health, and labour conditions of early modern maritime workers operating under extreme environmental constraints [43–50].

At the same time, accelerating permafrost degradation raises a critical question: not only what material is being lost, but which dimensions of past human experience risk disappearing before they can be adequately documented [51]. Whaling

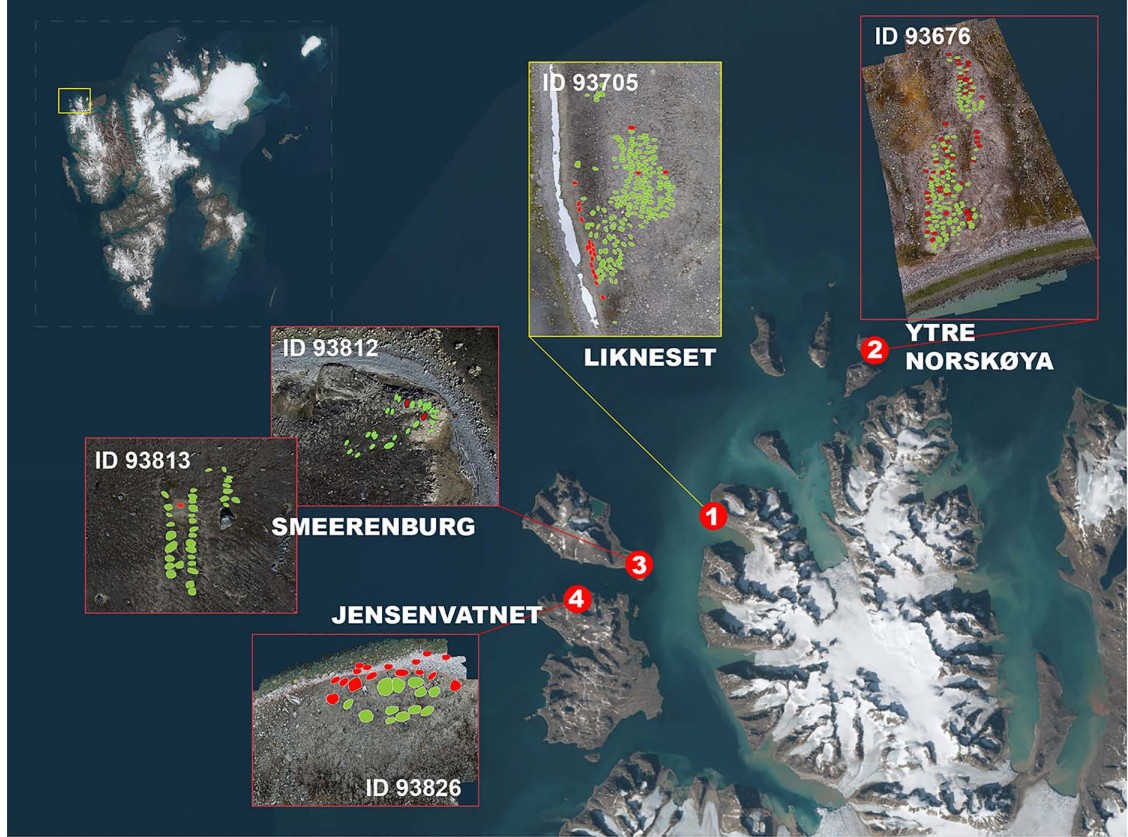

**Fig 2. Early modern whaling burial sites in the Smeerenburgfjorden region, northwestern Svalbard.** Map showing the distribution of early modern (17th–18th century) whaling burial sites in the Smeerenburgfjorden region, presented within the broader geographical context of the Svalbard archipelago. Insets show detailed orthophotos and site IDs of the largest and best-documented burial sites: (1) Likneset (225 registered graves), (2) Ytre Norskøya/Outer Norwegian Island (185 graves), (3) Smeerenburg (110 graves), and (4) Jensenvatnet on Danskøya/Danish Island (55 graves), illustrating the concentration of whaling-related burials along the fjord system. Red dots indicate excavated graves, while green dots represent undisturbed graves. Figure by Lise Loktu (NIKU). Basemap: TopoSvalbard, Norwegian Polar Institute. Orthophotos by Lise Loktu, Tommy Dahl Markussen, Espen Olsen, and Trygve S. Røysland, the Governor of Svalbard.

burial sites therefore function both as cultural heritage assets and as biological and environmental archives sensitive to climate-driven change.

The present analysis [52] focuses on Likneset ("Corpse Point") (ID 93705) [53], the largest known whaling burial site on Svalbard. The site has been excavated repeatedly over more than three decades [46,36], providing a rare opportunity to examine both preservation change and human skeletal evidence through time and across contrasting burial environments within a single site. This investigation forms the first in a planned series [52] of reanalyses of whaling burial sites from the Smeerenburgfjorden region, including all major sites shown in Fig 2.

### Aims and research questions

This study provides the first integrated taphonomic and osteological analysis of the Likneset burial site. The material comprises 20 individuals excavated across three phases (1985–1990, 2016, and 2019), of which 19 are included in the osteological analyses. The repeated excavation of closely spaced graves allows qualitative comparisons of preservation

                                                                

conditions across time under broadly comparable local environmental settings, as well as between erosion-exposed and geomorphologically stable areas within the burial site.

The analysis addresses two interrelated research questions. First, it examines how climate-driven environmental change, in combination with site-specific geomorphological conditions, has affected burial structures, coffins, skeletal remains, and textiles at Likneset. Attention is given to spatial contrasts within the burial site and to temporal change documented across excavation phases, to assess how permafrost thaw, active-layer deepening, and coastal erosion shape patterns of preservation and material loss. These observations are discussed in relation to broader Arctic preservation trends and their implications for cultural heritage management on Svalbard.

Second, the analysis investigates what osteological evidence from the Likneset burials reveals about the embodied health costs of labour among seventeenth- and eighteenth-century whalers operating in the High Arctic. By integrating indicators of metabolic disease, developmental stress, musculoskeletal degeneration, trauma, and dental pathology, the study explores how skeletal data can document lived experience within an early modern extractive system characterised by extreme environmental exposure, sustained physical labour, and limited nutritional resilience.

By explicitly linking biological data with taphonomic observations and environmental context, the results demonstrate how early modern whaling burial sites can be approached both as archives of labour and social history and as sensitive indicators of climate-driven cultural heritage degradation. In doing so, they highlight the value of archaeological datasets for advancing interdisciplinary research and informing evidence-based heritage management in rapidly changing Arctic environments.

## Background

### Climate change and consequences for Arctic cultural heritage

Climate change is increasingly recognised as a major driver of degradation affecting archaeological and built cultural heritage in the Arctic and sub-Arctic regions [2–6,9]. A key consequence of Arctic warming is permafrost thaw, which destabilises previously frozen sediments, alters local hydrology, and increases sediment mobility [57–61]. In periglacial environments such as Svalbard [19], these changes directly affect the preservation and integrity of archaeological contexts [21–24,36,51].

Permafrost thaw disrupts archaeological deposits through a range of thaw-driven processes, including subsidence, solifluction, retrogressive thaw slumps, thermo-erosion gullies, and downslope sediment movement [2–6,21–24,57–61]. Such processes may deform stratigraphy, displace features, and expose previously sealed deposits [2–5]. In burial contexts, ground instability can result in collapsed grave structures, coffin deformation, skeletal damage, and loss of spatial integrity [45,46,48], with direct implications for archaeological interpretation and comparability.

Coastal erosion represents a particularly severe and accelerating threat on Svalbard, especially in exposed fjord and shoreline settings where permafrost degradation interacts with wave action, storm surges, and reduced sea-ice cover [21–24,27,62–64]. Archaeological sites located on low-lying coastal terraces and moraine ridges, such as many early modern whaling burial sites, are therefore highly vulnerable to rapid truncation and loss over relatively short timescales.

Beyond mechanical disturbance, climate change also alters the microenvironmental conditions governing material preservation [2]. Rising temperatures increase active-layer thickness and extend the seasonal thaw period, allowing moisture and oxygen to penetrate deposits that were previously frozen and anoxic [4,5]. These conditions promote microbial activity and oxidation, accelerating the decay of organic materials such as wood, textiles, and bone [14–19,65,66]. Experimental and field-based studies from Greenland demonstrate strong temperature sensitivity in degradation rates, particularly under repeated wet–dry and freeze–thaw cycles [12–15,67,68].

In addition to climatic drivers, human activity, including tourism and research traffic, can exacerbate degradation at vulnerable sites through trampling, sediment disturbance, and direct damage to exposed features [25,26,28–32]. While

access restrictions have reduced direct anthropogenic impact at some locations, including Likneset [69], human pressure remains an important contextual factor when comparing preservation trajectories across sites.

### Historical context: Labour and working conditions in Arctic whaling

Early modern whaling in Svalbard developed rapidly during the seventeenth century as one of the earliest large-scale European extractive industries in the High Arctic, driven by intense competition among the dominant maritime powers of northwestern Europe [37–42]. From its inception, whaling relied on seasonal maritime labour operating under extreme environmental conditions, high physical demands, and substantial health risks [43,70–72]. Crews were recruited from across northwestern Europe and typically consisted exclusively of adult men engaged in highly specialised and physically demanding tasks related to hunting, processing, and shipboard labour [40–42,44].

Whaling developed from an early shore-based phase (c. 1612–1650) to a predominantly pelagic phase (c. 1650–1780), with processing shifting from land to sea [39–41]. Although organisational structures and recruitment patterns changed over time, Arctic whaling consistently depended on a highly mobile workforce characterised by repeated long-distance voyages, prolonged exposure to cold and wet conditions, and limited access to fresh food [43,71]. Mortality during whaling campaigns was significant, and deaths occurring in the Arctic were generally managed locally, resulting in the establishment of burial grounds along fjords and anchorages used by whaling fleets [36,43,50].

Most early modern whaling burial sites in Smeerenburgfjorden are associated with the period when pelagic whaling had become dominant [43,46], although precise chronological attribution remains incomplete. Nevertheless, sheltered fjord systems such as Smeerenburgfjorden retained logistical importance as seasonal hubs for anchorage, freshwater access, repair, and burial of deceased crew members [38, 39]. Burial practices were shaped primarily by practical constraints combined with broadly shared Christian funerary conventions, rather than by strict national traditions [43,46].

Historical and archaeological research demonstrates that whaling crews were socially and ethnogeographically diverse, reflecting transnational recruitment and labour mobility rather than homogeneous national groups [39–41,43,46,73]. Material culture recovered from burials, including clothing, personal items, and coffin construction, likewise reflects broadly shared maritime practices and pan-European material traditions [46]. These patterns caution against attributing individual burials to specific national enterprises and instead support labour-oriented and life-history approaches to interpretation.

## Studied site, materials and methods

### The Likneset burial site

Likneset is located on the eastern shore near the northern outlet of Smeerenburgfjorden, close to the terminus of the Kennedybreen glacier (Fig 3). The burial site is situated on a slightly elevated sediment surface approximately 10 m above present sea level, interpreted as a post-glacial deposit formed during deglaciation [74, 75]. Similar elevated sediment mounds hosting whaling burials are documented at Jensenvatnet [76], Ytre Norskøya, and Smeerenburg, suggesting shared geomorphological criteria in burial site selection.

Graves are generally shallow (c. 0.2–1.0 m below surface) and marked by small stone cairns, originally fitted with wooden crosses. Burial orientation varies (east–west and north–south), consistent with prolonged and episodic use of the burial site [46]. Radiocarbon dating links the site to seventeenth–eighteenth century pelagic whaling activity [77, 78]. Owing to its exposed coastal setting, Likneset is highly vulnerable to wave erosion, ice scouring, permafrost thaw, and sediment instability, resulting in progressive loss of burial contexts [79–81].

### Excavation history and material

Systematic documentation of Likneset began in the early 1980s, when at least 225 graves were recorded [46]. Parts of the burial site had already been lost to coastal erosion prior to excavation, and shoreline retreat has continued since [79–81].

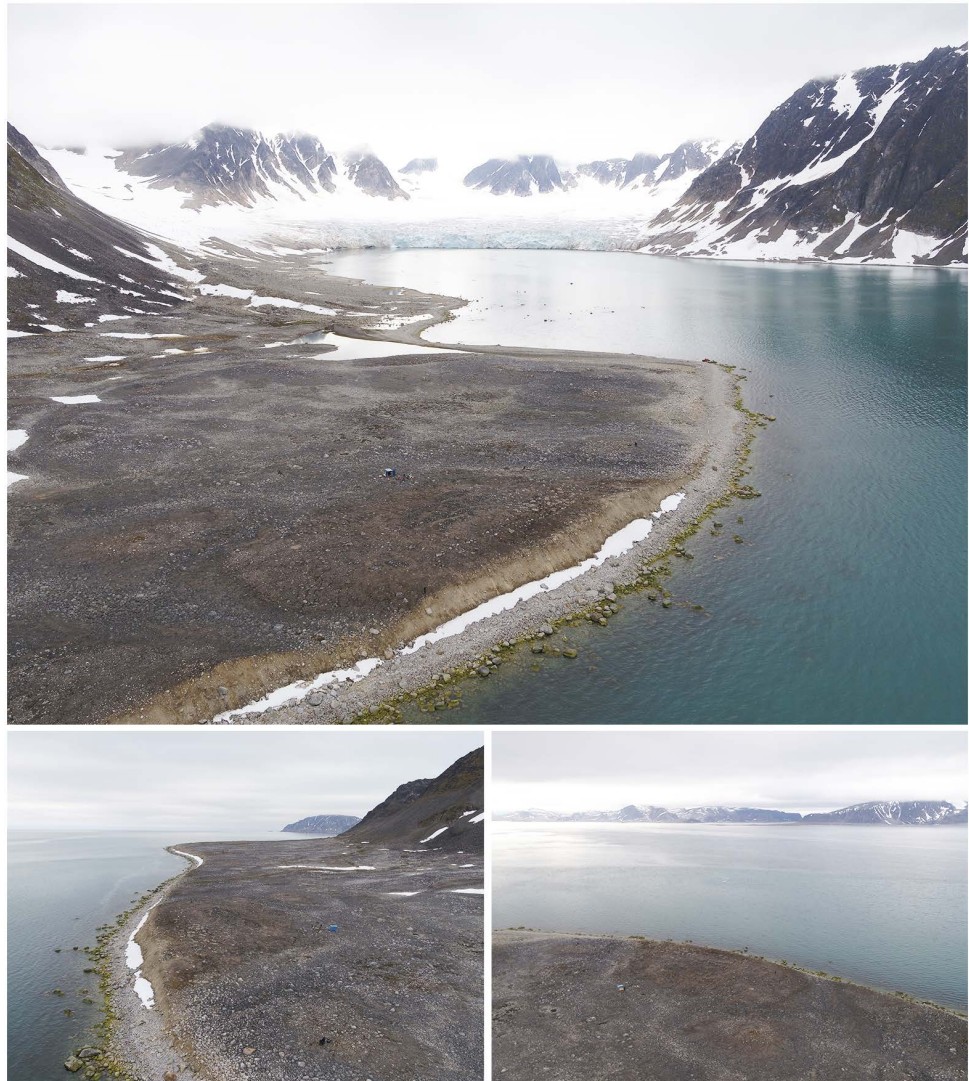

**Fig 3. The Likneset burial site and surrounding landscape, Smeerenburgfjorden, northwestern Svalbard.** Aerial views of the Likneset whaling burial site (ID 93705), located on the eastern shore near the northern outlet of Smeerenburgfjorden, facing the open Arctic Ocean to the northwest. The burial site is situated on a slightly elevated sediment surface likely formed during post-glacial deglaciation. This geomorphological setting contributes to the site's high vulnerability to coastal erosion, permafrost thaw, and related geomorphological processes. Photos by Lise Loktu and Espen Olsen, the Governor of Svalbard.

Twenty graves were excavated across three phases (Fig 4): Phase I (1985–1990, n = 14), Phase II (2016, n = 3), and Phase III (2019, n = 3) (Fig 4). Phase I graves were excavated from the erosion-exposed area, primarily *in situ*, with selected partial block lifts used to secure fragile materials [46]. Phase II graves, excavated from the same erosion-affected area, were recovered as block lifts and excavated under controlled laboratory conditions at the Svalbard Museum [54]. Phase III targeted graves from a more central, geomorphologically stable area and were recovered as intact block lifts and excavated at the museum [56].

Nineteen individuals are included in the osteological analyses; one individual from Phase I (Grave 211) could not be found in the curated collection. All available contextual information for this grave is nevertheless included in the analysis.

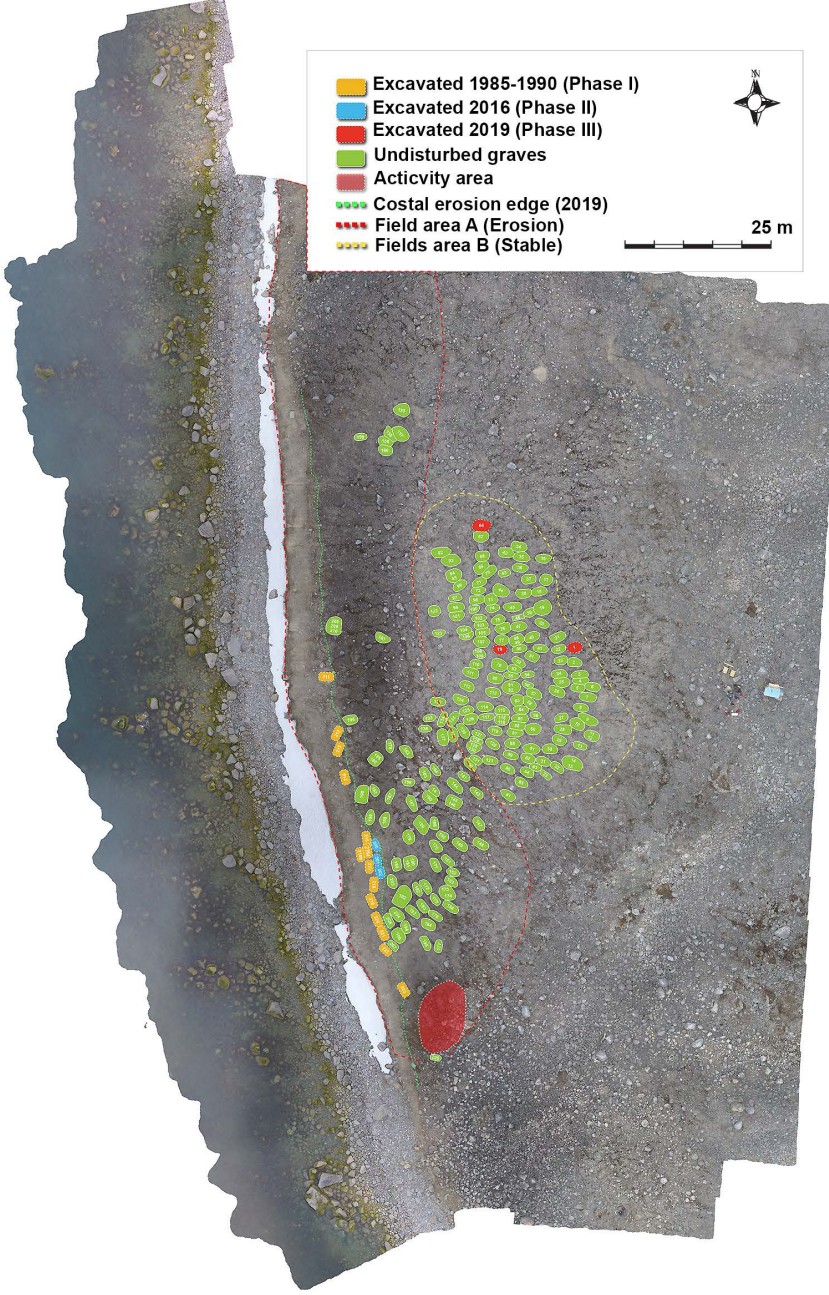

**Fig 4. Orthophoto of the Likneset burial site showing excavation phases, grave distribution, and erosion context.** Orthophoto illustrating the spatial distribution of recorded graves and excavated individuals across three excavation phases: Phase I (1985–1990), Phase II (2016), and Phase III (2019). Undisturbed graves are shown alongside excavated graves, with excavation phases indicated by colour. The dashed coastal line marks the documented erosion edge in 2019, highlighting the progressive loss of burial contexts due to shoreline retreat. Field area A denotes the erosion-affected marginal zone, while Field area B represents a more geomorphologically stable, inland area. Figure by Lise Loktu, NIKU. Orthophoto by Lise Loktu, Espen Olsen, and Trygve S. Røysland, the Governor of Svalbard.

Almost all skeletal remains and associated artefacts are curated at the Svalbard Museum, Longyearbyen. An overview of repository and specimen numbers, excavation history, and osteological analysis overview from the Likneset burial site is provided in S1 Table.

## Biomolecular analyses

Radiocarbon dating [77, 78], ancient DNA analysis [82,83], and stable isotope analyses [84] were conducted on selected individuals during the 2016 (n = 3) [54] and 2019 (n = 3) [56] excavations, supplemented by comparative material from Smeerenburg in 2017 (n = 2) [55]. In the present study, biomolecular data are used selectively to support chronological attribution, biological sex determination, and broader contextual interpretation. Detailed genetic and isotopic results are reserved for forthcoming dedicated publications. An overview of available biomolecular data is provided in S2 Appendix B.

## Analytic framework and preservation assessment

**Erosion data and environmental monitoring.** Environmental assessment is based on field observations, excavation records, shoreline measurements [79, 80], and drone-based orthophotography acquired since 2016 [81]. These datasets provide qualitative and semi-quantitative records of shoreline position and burial exposure but lack consistent georeferencing. Comparison between early documentation and recent orthophotos indicates shoreline retreat of approximately 2.2 m over c. 30 years along parts of the erosion front [54].

**Geomorphological variability.** Although no site-specific geomorphological mapping or long-term permafrost temperature data are available, aerial imagery and field observations document pronounced small-scale variability across the sediment mound hosting the burial site (Fig 5). The mound shows surface cracking and incipient deformation along its outer margins, while the central area remains comparatively intact. The most pronounced cracking and surface disruption occur along the actively eroding coastal edge, where darker sediment coloration indicates higher moisture content and ongoing thaw-related processes.

Based on these observations, two analytical field areas were defined:

**Field area A,** an erosion-affected coastal margin characterised by surface cracking, sediment instability, and downslope movement; sediments consist of poorly consolidated sand and silt with interspersed gravel and stones.

**Field area B,** a more central and geomorphologically stable area with intact cairns and burial structures; sediments appear finer-grained and more compact, with better internal coherence and less visible surface cracking, indicating reduced exposure to active erosional and thaw-related processes.

**Analytic framework.** To ensure comparability across excavation phases and burial micro-environments, all osteological, taphonomic, and contextual data were reassessed using a unified analytic framework. Field documentation from the 1980s was systematically re-evaluated using the same criteria applied to material excavated in 2016 and 2019.

Individuals were grouped according to both excavation phase and environmental setting. **Group A** (coastal/erosion-affected) includes individuals excavated during Phase I (1985–1990, n = 14) and Phase II (2016, n = 3), while **Group B** (central/geomorphologically stable) comprises individuals excavated during Phase III (2019, n = 3).

**Preservation assessment.** Preservation was recorded using a semi-quantitative scoring system encompassing burial structure integrity (A1–A4), coffin integrity (B1–B4), skeletal completeness (C1–C3), skeletal preservation (S1–S4), and textile preservation (T1–T4) (Table 1). Environmental indicators (sediment type, drainage, moisture, permafrost-related disturbance) were recorded descriptively. Detailed analysis of textile conditions will be presented in a separate study; here, only the presence and relative amount of textile material are recorded for comparative purposes. Scoring tables and quantitative summaries are provided in S1 Appendix A.

## Osteological methods

Osteological analyses were conducted at the Svalbard Museum laboratory between 2016 and 2023 [51,86,87], following internationally recognised standards [88–92]. All assessments were conducted by the same analyst (E.T. Brødholt). Skeletal material excavated in the 1980s was re-analysed to ensure methodological consistency across the assemblage.

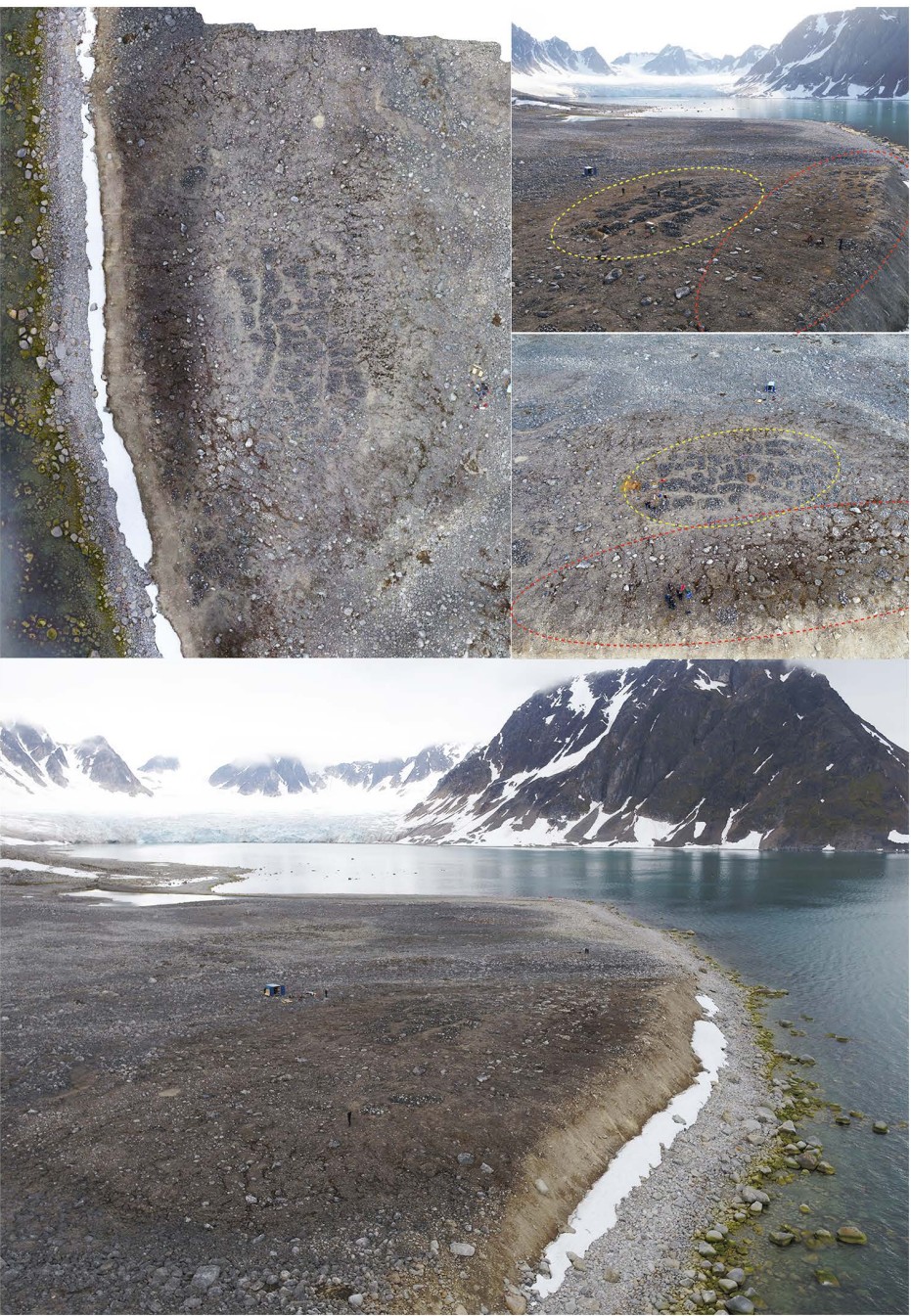

**Fig 5. Geomorphological setting of the Likneset burial site.** Drone-based aerial imagery illustrating small-scale geomorphological variability across the sediment mound hosting the burial site. The central area of the mound appears comparatively intact, while the outer margins show extensive surface cracking, darker sediment coloration, and incipient deformation, particularly along the actively eroding coastal edge. Yellow dashed outlines mark comparatively stable areas (Field area B), whereas red dashed outlines indicate erosion-affected zones (Field area A). Figure by Lise Loktu, NIKU. Photos by Lise Loktu and Espen Olsen, the Governor of Svalbard.

**Table 1. Variables and semi-quantitative scoring system applied to assess preservation conditions of burial structures, coffins, skeletal remains, and textiles at Likneset.**

| Category | Recorded Variables | Scoring System |
|---|---|---|
| **Burial Structure (A1–A4)** | Condition of stone cairn; displacement; erosion impact | A1: Intact stone cairn with no displacement. A2: Disturbed cairn; coffin may be visible; minor downslope movement. A3: Heavily disturbed or partially destroyed cairn; coffin and/or skeleton displaced; clear downslope movement. A4: Complete destruction of the grave context; cairn, coffin, and skeletal elements displaced and scattered downslope. |
| **Coffin Integrity (B1–B4)** | Structural preservation of coffin boards, lid, and base | B1: Intact coffin; lid and sideboards preserved in their original position. B2: Partially collapsed coffin with minor deformation; overall structural outline intact. B3: Major collapse; lid pressed downward; sideboards displaced; coffin shape distorted. B4: Complete structural failure; skeletal elements and associated materials displaced and scattered downslope. |
| **Skeletal Completeness (C1–C3)** | Proportion of anatomical elements present | C1: >90% of skeleton present. C2: 50–90% present. C3: <50% present. |
| **Skeletal Preservation (S1–S4)** | Surface condition; erosion; fragmentation | S1: Intact cortical bone surfaces with minimal alteration; preservation of soft tissues (e.g., hair, skin fragments, nails, cartilage, and possible internal tissue masses). S2: Minor cracking or flaking of cortical bone surfaces and/or minor damage to articular surfaces. S3: Moderate cracking or flaking of cortical bone surfaces and/or moderate damage to articular surfaces. S4: Extensive cracking and flaking of cortical bone surfaces and/or severe damage to articular surfaces. |
| **Textile Preservation (T1–T4)** | Quantity, integrity, and recognisability of textile remains | T1: Multiple complete or near-complete garments preserved; diagnostic features intact. T2: Several identifiable fragments; at least one partially preserved garment. T3: Small fragments only; minimal diagnostic value. T4: No preserved textile material. |
| **Coffin Fill/ Sealing Materials** | Presence and type (e.g., sawdust, wood shavings, organic packing) | Descriptive only (no scoring). |
| **Environmental Indicators** | Sediment type; drainage; moisture conditions; permafrost-related disturbances | Descriptive only (no scoring). |

The categories were developed in collaboration with conservators and curators at the Svalbard Museum and are based on the curatorial database Primus and museum guidelines for condition assessment [85].

Sex estimation was based primarily on pelvic morphology [88], supplemented by cranial traits [91]. Supporting evidence was derived from the overall size and robusticity of post-cranial elements and the degree of development of muscle attachment sites. Age at death was estimated using multiple indicators, including pubic symphysis morphology [93,94], auricular surface, cranial suture closure [95], rib ends, dental wear, and degenerative changes. Stature was estimated from femoral length using regression equations by Trotter and Gleser [96], with Sjøvold's method applied where sex estimation was uncertain [97].

Pathological assessment focused on indicators relevant to labour and lived experience, including degenerative joint disease, trauma, musculoskeletal stress markers, enthesopathies, metabolic disease, and nutritional stress [43,98–101]. Dental pathology and activity-related wear, including pipe-smoking facets, were documented following Buikstra and Ubelaker [88]. Detailed osteological data are provided in S2 Appendix B.

**Ethical considerations and permits**

No permits were required for the described study, which complied with all relevant regulations. Excavations were authorised under the Svalbard Environmental Protection Act [102] and approved by the Norwegian National Committee for Research Ethics on Human Remains [103] (REC No. 2017/153).

## Results

### RQ1: Preservation and taphonomic patterns

Burial preservation at Likneset displays marked spatial and temporal variation linked to grave location, excavation phase, and local geomorphological conditions. Differences are observed across all analytical levels, including burial structure, coffin integrity, skeletal condition, and textile preservation.

**Burial structure and spatial setting.** Phase I burials (1985–1990) from Field area A (n = 14) exhibit extensive structural disturbance associated with active erosional processes, including solifluction, subsurface instability, and downslope movement of sediments and stones (Fig 6). Most graves are classified as heavily disturbed (A3), reflecting collapsed cairns, displaced stone linings, and deformation of coffin structures. In several cases, coffin lids had collapsed and sideboards were displaced, resulting in partial disturbance of skeletal remains and textiles. One grave (Grave 214) is classified as completely destroyed (A4), with coffin elements and skeletal remains dispersed downslope. No Phase I graves were classified as intact (A1).

Phase II burials (2016), also located in Field area A (n = 3), show a comparable but less advanced degree of structural disturbance (Fig 7). All three graves are classified as moderately disturbed (A2), indicating largely preserved but structurally compromised burial contexts. Stone cairns are better preserved than in Phase I, likely reflecting shorter cumulative exposure to sediment movement. All coffins exhibit collapsed lids and evidence of sediment infiltration.

In contrast, Phase III burials (2019) from Field area B, located in the more central and geomorphologically stable part of the burial site (n = 3), display substantially better preservation (Fig 8). Two graves (Graves 1 and 78) are classified as intact (A1), while one grave (Grave 66) is classified as moderately disturbed (A2) due to a collapsed coffin lid and minor sediment intrusion.

**Coffin integrity and coffin fill.** In Phase I (n = 14), most coffins show advanced structural failure, with 40% classified as collapsed (B3) and a further 25% as partially collapsed (B2). One coffin (Grave 214) is classified as completely destroyed (B4). Phase II coffins (n = 3) are consistently classified as B2, representing intermediate preservation. In Phase III (n = 3), coffin preservation is markedly improved, with two intact coffins (B1) and one nearly intact coffin (B2), where lid collapse is attributed to overlying stone load rather than sediment instability.

Across all phases, coffin fill is dominated by sawdust and wood shavings, typically used both as a basal fill and body covering. Mixed fills of coarse shavings and sawn wood fragments are also documented. One grave (Grave 223) contained grass or moss beneath the head, while three Phase I graves lack documented coffin fill; notably, these rank among the best-preserved Phase I burials in terms of textile preservation. A single grave from Field area B (Grave 1) contained buckwheat chaff completely covering the body. This represents the only occurrence of buckwheat chaff at Likneset and is otherwise documented only in two graves at Ytre Norskøya [43].

**Skeletal preservation and completeness.** Skeletal preservation closely mirrors burial context conditions. Poorer preservation is associated with erosion exposure, coffin collapse, and prolonged post-mortem disturbance, including moisture infiltration and mechanical stress.

Skeletons from Field area A (Phases I–II) show the greatest variation in preservation but are predominantly well preserved. Among Phase I skeletons (n = 13), 42% are classified as S1, with the remainder falling within categories S2–S3. Phase II skeletons display a comparable pattern, although the small sample size precludes statistical comparison.

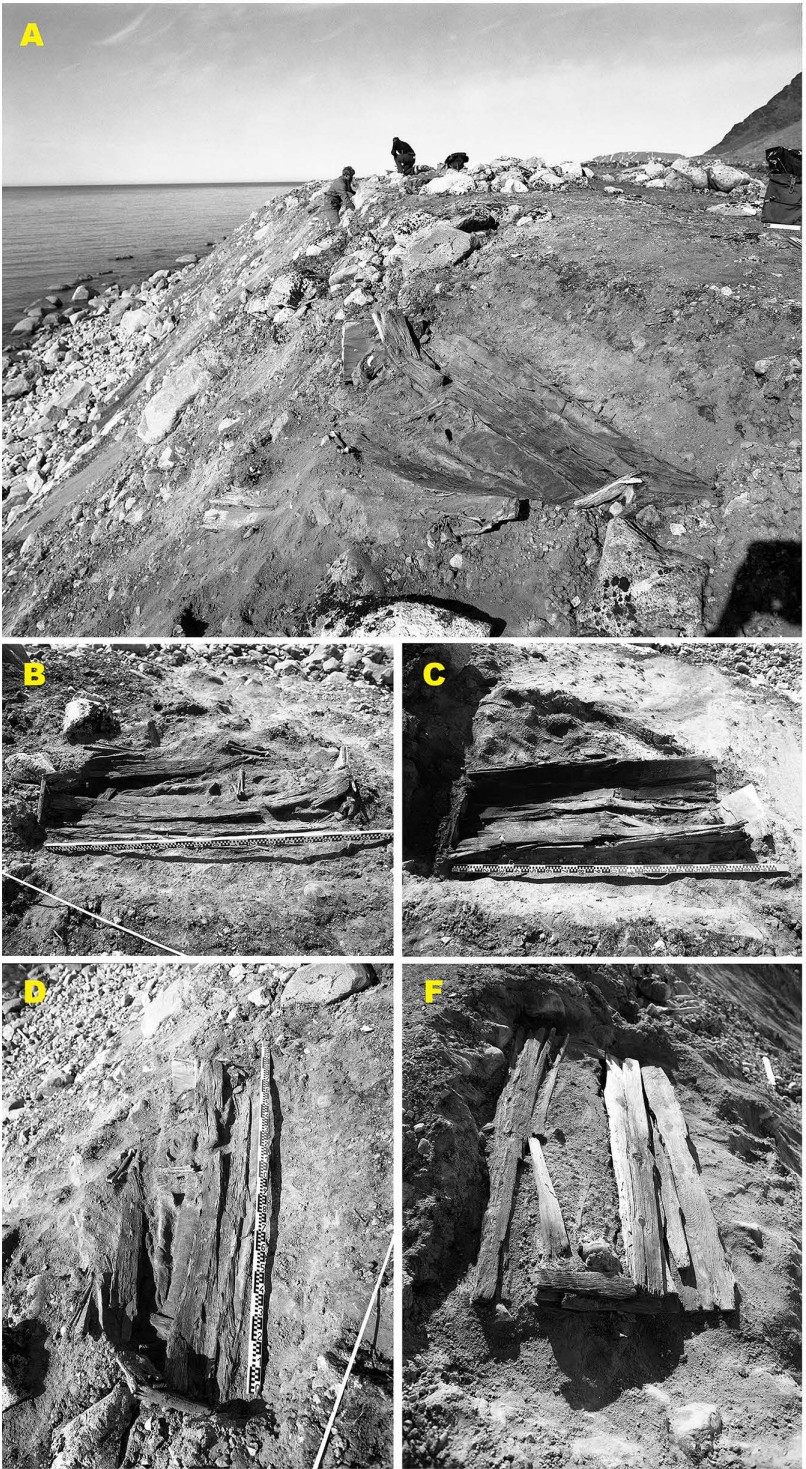

**Fig 6. Erosion-related damage to graves at Likneset documented during the 1985 excavations (Phase I).** Representative graves from the erosion-exposed coastal zone (Field area A), including Grave 220 (A), Grave 219 (B), Grave 217 (C), Grave 214 (D), and Grave 215 (E), illustrating varying degrees of disturbance caused by subsurface instability, solifluction, and ground fissuring. Stone cairns and linings have shifted downslope, and coffins show collapsed lids, warped boards, and displacement of structural elements. Such disturbance alters burial microenvironments by enabling the infiltration of moisture, sediment, and oxygen, thereby accelerating post-burial degradation processes. Photos by Dag Nævestad, Tromsø Museum.

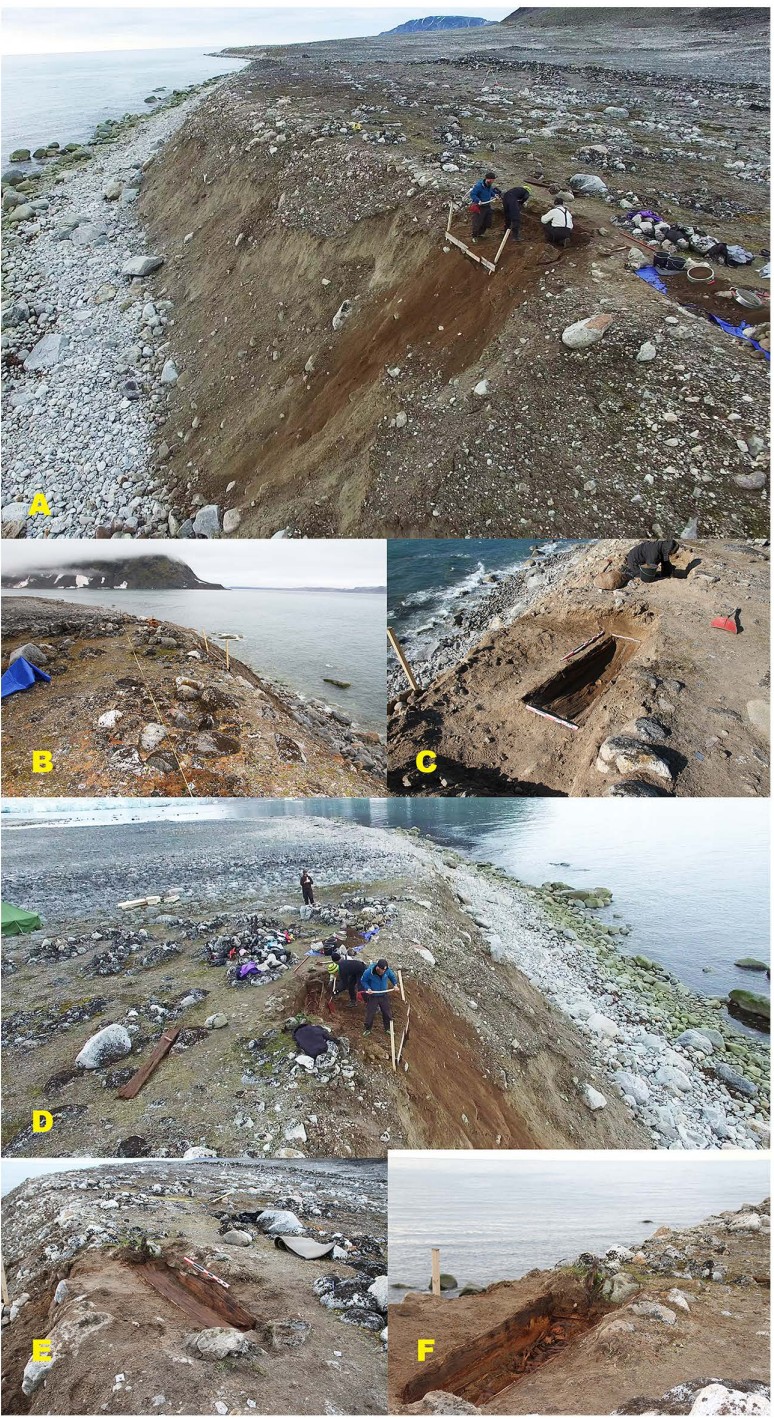

**Fig 7. Graves excavated in 2016 (Phase II) from the erosion-exposed coastal zone (Field area A).** Overview and excavation sequence illustrating moderately disturbed burial contexts (A2) under active erosion. (A) Overview of the 2016 excavation area with Grave 200 under excavation along the actively eroding coastal edge. (B–C) Grave 201 before and during excavation, showing surface exposure, collapsed coffin lid, and sediment infiltration. (D) Overview of the excavation of Grave 202 within the same erosion-affected zone. (E–F) Grave 202 before and after excavation, documenting a largely preserved but structurally compromised coffin with lid collapse and downslope sediment movement. Photos by Arild Vivås and Snorre Haukalid, the Governor of Svalbard.

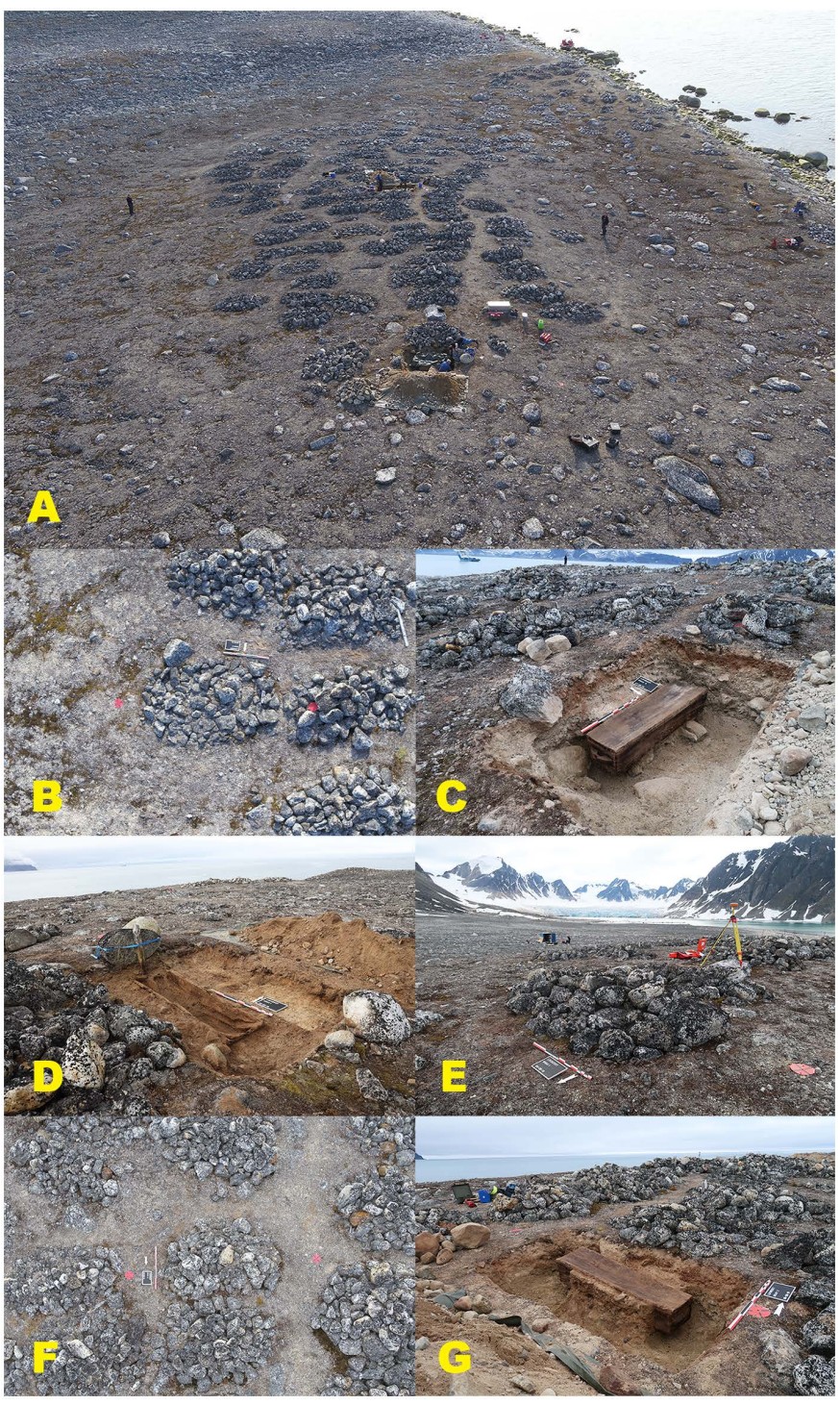

**Fig 8. Graves excavated in 2019 (Phase III) from the central, geomorphologically stable area (Field area B).** Overview and excavation sequence illustrating improved preservation conditions compared to the erosion-exposed coastal zone. (A) Aerial overview of the central part of the burial site, showing a dense concentration of intact stone cairns and limited surface disturbance. (B–C) Grave 1 before and after excavation, an intact stone cairn (A1) and a well-preserved coffin structure (B1). (D–E) Grave 66 before and after excavation, showing an intact stone cairn (A1) with a collapsed coffin lid (B2). (F–G) Grave 78 before and after excavation, likewise exhibiting intact burial architecture and coffin (A1/B1). The graves are situated in finer-grained, more compact sediments with minimal surface cracking, reflecting reduced exposure to erosion and thaw-related geomorphological processes. Photos by Lise Loktu and Espen Olsen, the Governor of Svalbard.

Approximately two-thirds of Field area A burials exhibit post-mortem alterations consistent with erosion-related processes, including cranial crushing, cortical surface erosion, staining, moss growth, joint surface degradation, and cracking or flaking. Some fragmentation in Phase II material is also attributed to post-excavation handling and storage.

In contrast, Phase III skeletons from Field area B show consistently excellent preservation (S1), with intact articulation and minimal post-mortem disturbance, aside from some post-excavation cortical surface cracking.

Despite variation in preservation quality, skeletal completeness is high across the assemblage: all but one individual (Grave 214) are classified as nearly complete (C1). Soft tissues, including hair, skin fragments, fingernails, and internal organic masses, were observed across all excavation phases.

**Textile preservation.** Textile preservation follows the same spatial and temporal pattern observed for burial structures and skeletal remains and appears particularly sensitive to environmental disturbance.

Phase I burials from Field area A yield the highest number of recorded finds and recognisable garments (Fig 9), including woollen caps, jackets, trousers, knitted stockings, linen shirts, scarves or cravats, bedding elements, and, in one case, shoes. Approximately half of the individuals have more than five preserved garments, while two graves (211 and 214) contain no preserved textiles. Grave 218 yielded the highest number of finds (33 items, including 10 textile garments).

Phase II burials from the same erosion-exposed area show markedly poorer textile preservation (Fig 10). Only one grave yielded highly fragmentary textile remains, interpreted as remnants of stockings, while the remaining burials contained no preserved textiles despite the presence of buttons, indicating that garments were originally present at burial but subsequently lost.

Textiles from Phase III burials in Field area B are notably better preserved than those from Phase II and broadly comparable to the Phase I assemblage (Fig 11). Preserved garments include woollen trousers, a jacket, knitted stockings, caps, and, in one case, a finely woven linen shirt. Ongoing assessment suggests that overall textile preservation is somewhat less robust than in material excavated during the 1980s and requires further investigation [104]. Across all excavation phases, animal fibres – particularly wool – are consistently better preserved than plant-based materials such as linen, a pattern also documented at Ytre Norskøya and Jensenvatnet [49,50].

## RQ2: Demography and health indicators

The Likneset skeletons are generally well preserved compared to many European and global archaeological assemblages, allowing detailed assessment of pathological features.

**Biological sex and age at death.** All 19 individuals were assessed as biologically male. Sex determination is further supported by aDNA analysis of six individuals excavated during Phases II and III (Graves 200–202, 1, 66, and 78) [82,83].

Age-at-death estimates indicate a mortality profile dominated by young and middle adults. Thirteen individuals (68%) are classified as Young Adults (20–34 years), while six individuals (32%) fall within the Middle Adult category (35–49 years). One individual falls at the transition between these age categories. No juveniles, adolescents, or individuals older than 50 years are represented in the assemblage. Within the Young Adult group, six individuals (32% of the total assemblage) are more narrowly estimated to be between 20 and 25 years of age based on epiphyseal fusion, pubic symphysis morphology, and indicators of overall skeletal maturity, including the appearance of articular surfaces.

**Stature.** Stature estimates were obtained for all 19 individuals and range from 166.5 to 179.7 cm, yielding a total variation of 13.2 cm (mean = 173.3 cm; median = 172 cm). Stature values are broadly evenly distributed across age categories, with no pronounced differences between young and middle adults, and no extreme outliers. The observed stature range is consistent with earlier analyses of the Likneset assemblage [50].

**Dental pathology and wear.** Dental wear is predominantly mild to moderate across the assemblage. Minimal wear is recorded in two individuals, mild wear in seven individuals, moderate wear in seven individuals (37%), and pronounced wear in one individual. Wear generally increases with age, although marked variation is observed within age categories.

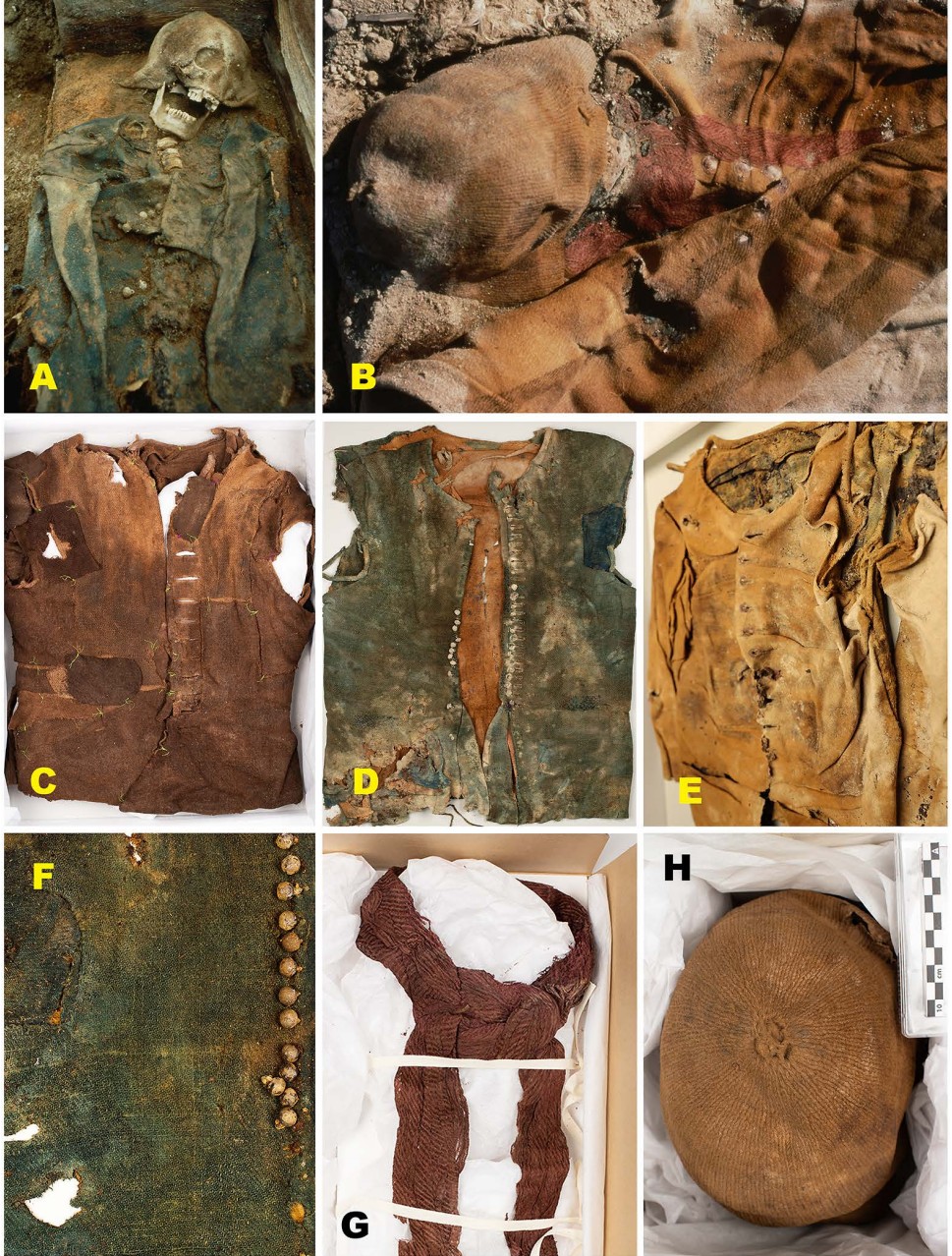

**Fig 9. Textile preservation in Phase I burials from Field area A.In Phase I, textile preservation is generally exceptionally good where present, although substantial variability is observed.** The figure illustrates examples of some of the best-preserved garments from (A) and (F) Grave 222; (B) and (G) Grave 216B; (C) and (D) Grave 218; and (E) and (H) Grave 216A. Preserved garments include woollen caps, jackets, trousers, knitted stockings, and bedding elements, reflecting favourable preservation conditions in several Phase I burials despite erosion exposure. The preserved textiles consist mainly of wool, with occasional remains of degraded linen shirts and trousers (likely undergarments), as well as silk scarves (cravats) recovered from the neck area in two graves (Graves 216A and 216B). Photos by Dag Nævestad, Tromsø Museum (A-B) and Lise Loktu, NIKU (C-H).

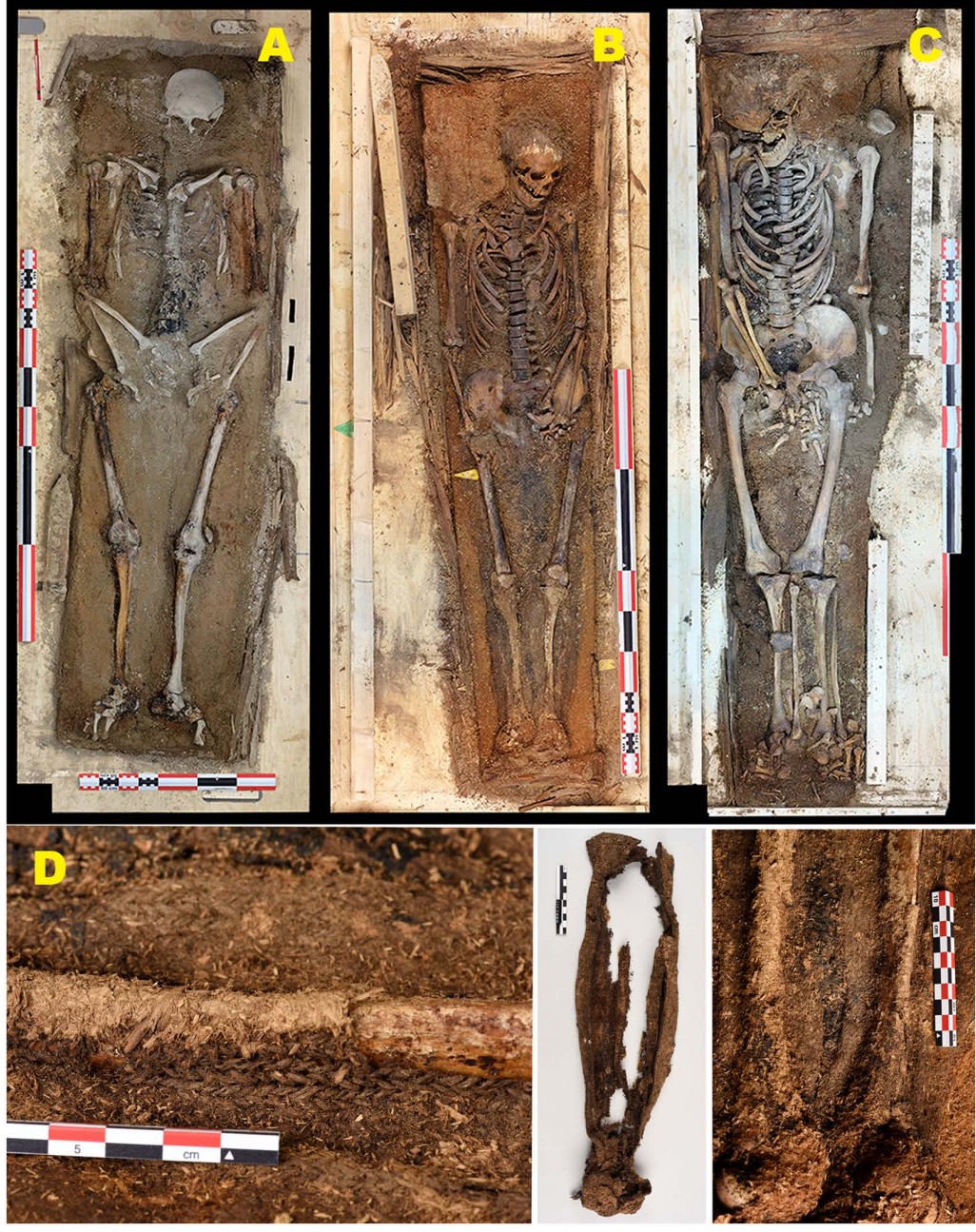

**Fig 10. Textile preservation in Phase II burials from Field area A.** Phase II burials show markedly poorer textile preservation compared to Phase I. The figure illustrates Graves 200 (A), 201 (B), and 202 (C). Only Grave 201 yielded textile remains, consisting of small, highly degraded wool fragments (D) interpreted as remnants of knitted stockings. No preserved textiles were recorded in Graves 200 or 202, despite the presence of buttons indicating that garments were originally present at burial. Photos and orthomosaics by Arild Vivås, the Governor of Svalbard.

Dental disease is present but limited. Dental caries and periapical abscesses are each recorded in seven individuals (37%), while periodontal disease is suggested in a smaller number of cases based on alveolar resorption and root exposure. Dental calculus is infrequently preserved, likely reflecting post-depositional loss rather than absence in life.

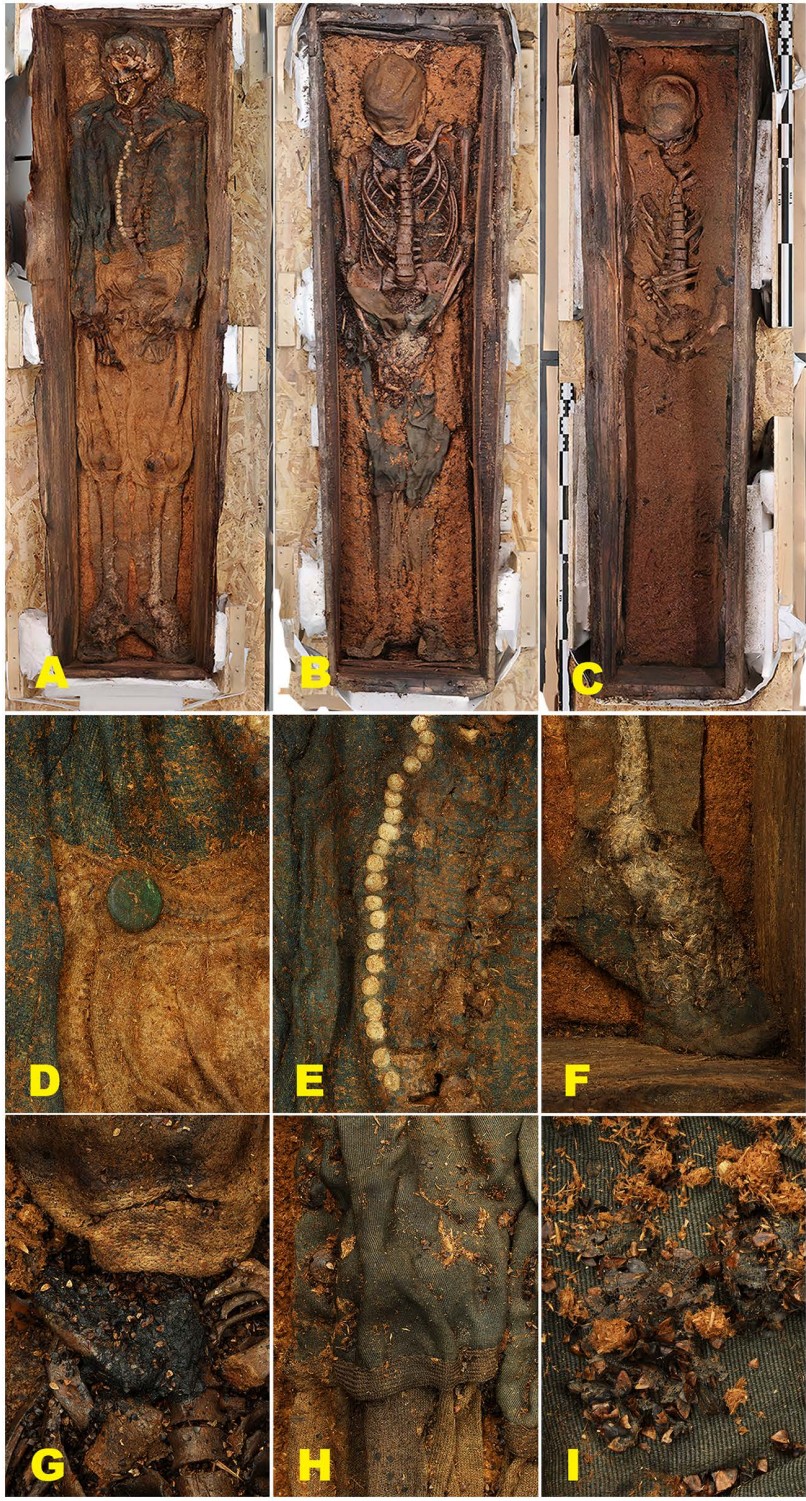

**Fig 11. Textile preservation in Phase III burials from Field area B.** Phase III burials from Field area B show markedly better textile preservation than Phase II, reflecting more stable burial conditions and reduced environmental disturbance. The figure illustrates Grave 1 (A, D–F), Grave 66 (B, G–I), and Grave 78 (C). Preserved textiles include a woollen jacket (E); finely felted woollen stockings (F); a very finely woven pair of woollen trousers (H); fragmentary remains of a blue-striped linen shirt (I); and a blue silk neck scarf (G, cravat). Overall, the textile assemblage from Phase III is broadly comparable to that of Phase I in terms of preservation quality, although the garments are generally in poorer structural condition. Grave 78 yielded only a single preserved garment, a woollen cap. Photos and orthomosaics (A-C) by Lise Loktu, the Governor of Svalbard and NIKU.

Distinct pipe-related wear facets attributable to clay pipe smoking are observed in 15 of the 19 individuals (79%). These typically occur as bilateral notches between the lateral incisors and canines or between canines and first premolars, affecting both the maxillary and mandibular dentition (Fig 12). In addition, several individuals exhibit pronounced anterior tooth wear exceeding that observed on posterior teeth, potentially reflecting habitual use of the anterior teeth as tools during work-related activities.

**Developmental stress markers.** Enamel hypoplasias are recorded in 12 of the 19 individuals (63%). Where present, the defects often affect multiple teeth within the same individual and are most frequently observed on the anterior dentition, particularly incisors and canines. The defects are predominantly linear in form.

The distribution and recurrence of enamel hypoplasias indicate repeated episodes of physiological stress during childhood and adolescence, broadly corresponding to the period of crown formation (approximately 2–15 years of age) [98,101]. No clear association is observed between the presence of enamel hypoplasias and adult stature. However, the coexistence of a high prevalence of hypoplasias with comparatively tall adult stature warrants further consideration and is addressed in the Discussion.

**Metabolic disease.** Skeletal indicators consistent with scurvy, caused by prolonged vitamin C deficiency [99, 100], are recorded in 18 of 19 individuals (95%). Observed features include dark discoloration of articular ends – particularly affecting the femora, tibiae, humeri, and fibulae – porous cortical bone surfaces, vascular impressions on tibial shafts, and periosteal reactions, all of which are consistent with recurrent haemorrhaging and chronic oedema (Fig 13) [98–100].

In more severe cases, generalised skeletal involvement and osteopenia suggest prolonged or recurrent metabolic stress associated with long-term malnutrition. Indicators of scurvy are present in both young and middle adults, although more extensive involvement is more frequently observed among older individuals. Skeletal changes consistent with rickets are identified in one individual, affecting tibial and fibular morphology and reflecting vitamin D deficiency during childhood. This represents the only clear case of rickets in the assemblage (Fig 13).

**Degenerative joint disease and activity-related stress.** Degenerative joint disease (DJD) and activity-related musculoskeletal stress markers are recorded in 18 of 19 individuals (95%). Changes affect both the axial and appendicular skeletons and are frequently multi-regional within individuals (Fig 14).

The upper body is most consistently involved. Degenerative and activity-related changes affecting the shoulders, clavicles, sternum, and elbows are documented in most individuals, particularly in the humeri, scapulae, clavicles, and sternum. In five individuals (26%), predominantly young adults, pronounced enthesopathic changes occur at the inferomedial clavicle at the attachment of the costoclavicular ligament.

Spinal involvement is recorded in 13 individuals (68%) and includes vertebral lipping, facet joint degeneration, Schmorl's nodes, and evidence of disc injury or prolapse. These changes occur in both young and middle adults and are notable given the predominantly young adult age profile of the assemblage.

Lower limb involvement is also common, affecting the hips, sacroiliac joints, knees, feet, and toes. Changes frequently co-occur with axial and upper-limb degeneration and include joint surface impressions, cystic activity, and enthesopathic alterations involving the femur, tibia, fibula, metatarsals, and phalanges. One individual also exhibits abnormal bone growth within the external auditory canal, consistent with external auditory exostoses [105].

Overall, the distribution of DJD and activity-related stress markers indicates widespread mechanical loading affecting the skeleton as a whole.

**Trauma and injuries.** Evidence of trauma is present but limited. Healed fractures are identified in two individuals, affecting the radius and foot bones. No unequivocal unhealed or perimortem fractures are recorded. Additional injury-related changes include tendon and ligament avulsions, joint surface damage, and spinal injuries affecting the shoulder girdle, spine, hips, and feet.

The predominance of healed injuries indicates survival after traumatic events and suggests that mortality within the assemblage was more closely related to cumulative physiological stress than to acute fatal trauma.

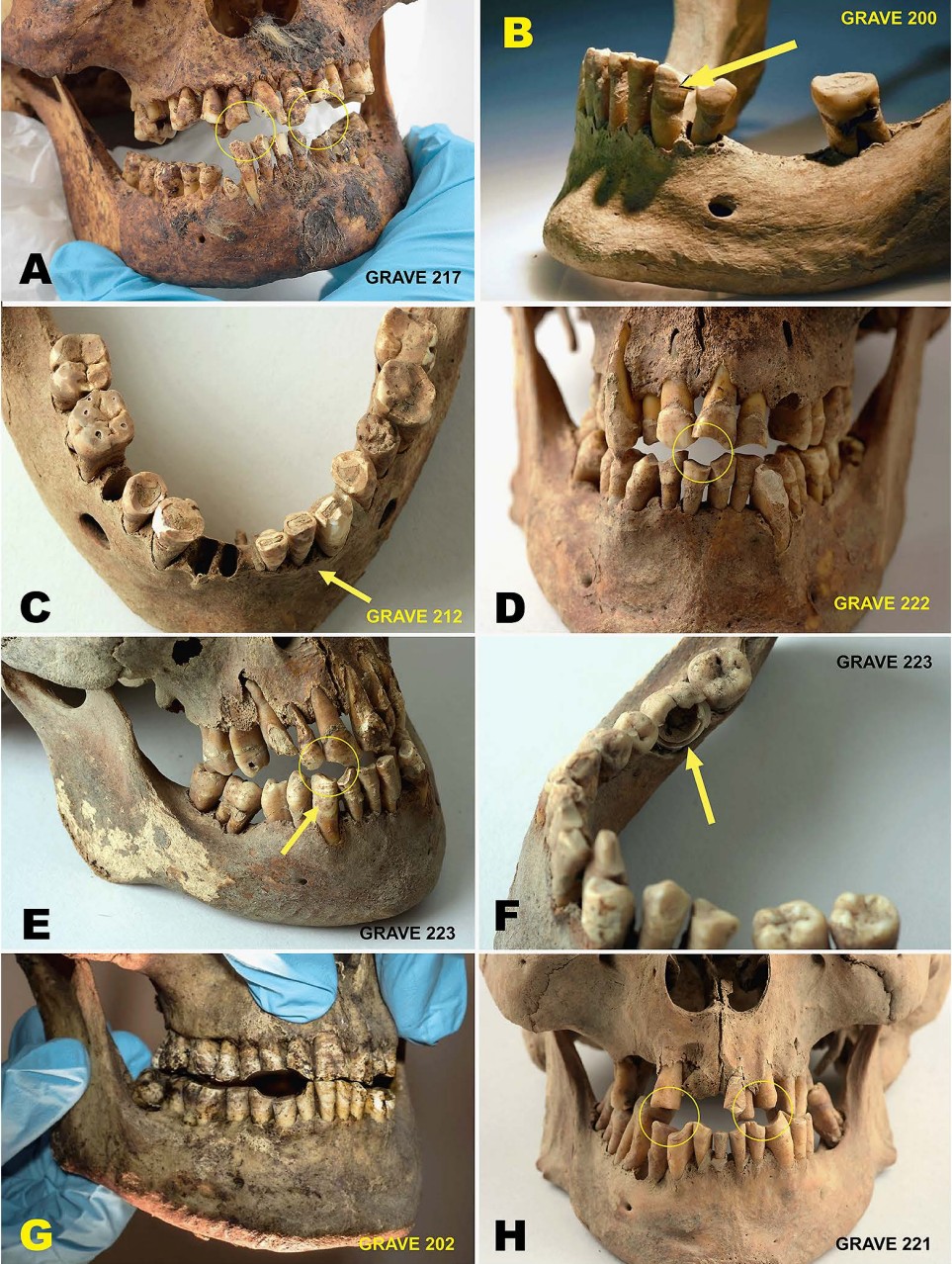

**Fig 12. Dental wear and pathology in the Likneset assemblage.** Examples of dental wear, pathology, and non-masticatory tooth use observed in individuals from Likneset. Panels A–H illustrate anterior tooth wear and characteristic bilateral notches between incisors, canines, and premolars consistent with habitual clay pipe smoking, recorded in 79% of individuals. Several individuals also exhibit pronounced anterior wear exceeding posterior wear, suggesting non-masticatory use of the teeth in work-related activities. Additional features include dental caries, periapical lesions, and alveolar bone changes indicative of inflammatory processes (e.g., panels C, F). Enamel hypoplasia, reflecting episodes of childhood physiological stress, is visible on multiple teeth in 63% of the individuals (e.g., panels B, E). Figure by Lise Loktu, NIKU. Photos by Lise Loktu, Elin T. Brødholt, and Carina V.S. Knudsen, the Governor of Svalbard and NIKU.

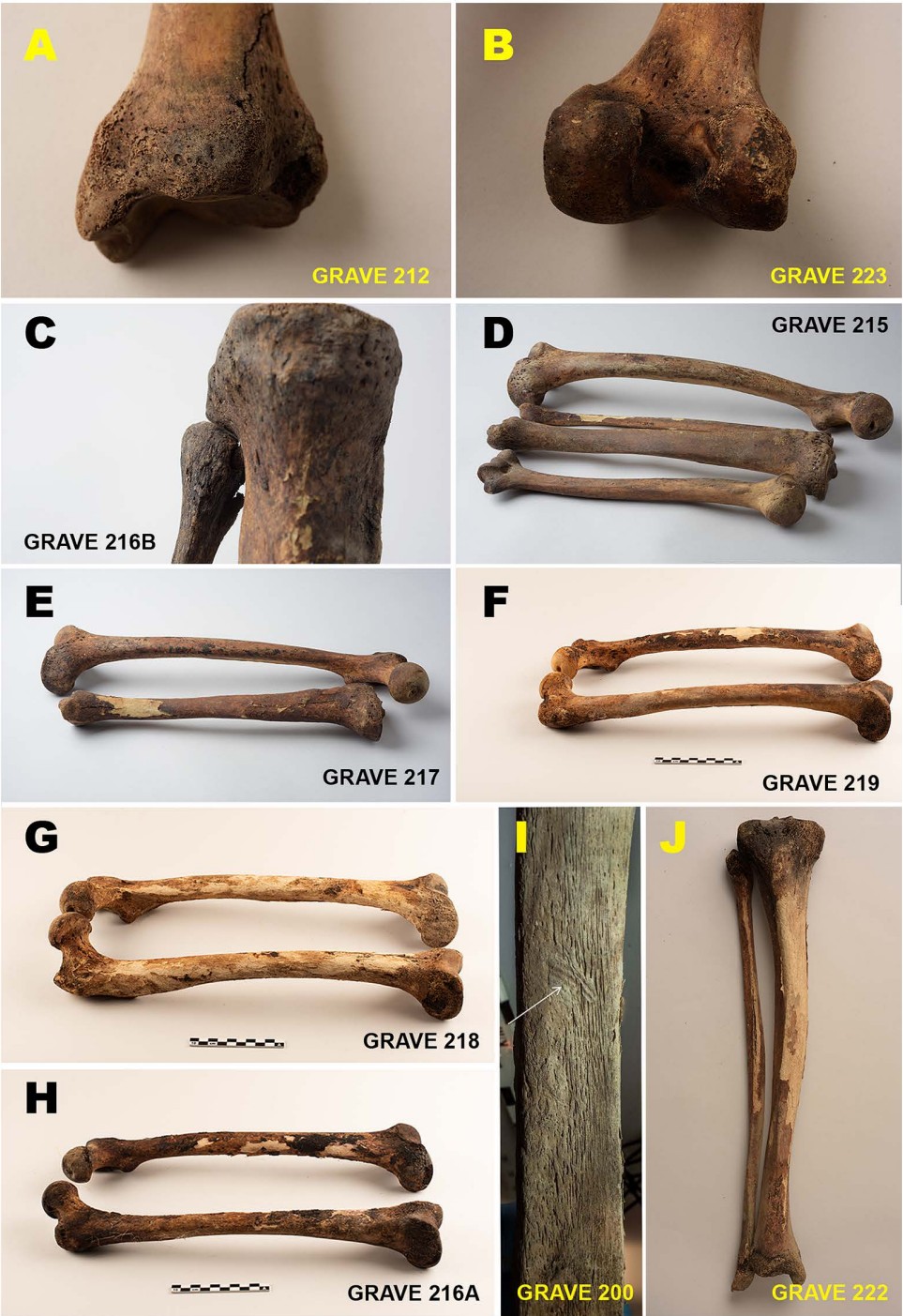

**Fig 13. Skeletal indicators consistent with scurvy in the Likneset assemblage.** Selected examples of postcranial skeletal changes associated with prolonged vitamin C deficiency (Panels A–J). The figure shows dark discoloration of joint ends, porous and irregular cortical surfaces, vascular impressions along tibial shafts, and periosteal reactions, with the femora, tibiae, humeri, and fibulae most prominently affected. The examples represent varying degrees of severity observed across the assemblage and reflect recurrent haemorrhaging and chronic oedema linked to long-term nutritional stress. Panel I shows a vascular impression on a tibial shaft consistent with scurvy, and Panel J illustrates a case of rickets. Figure by Lise Loktu, NIKU. Photos by Lise Loktu, Mikael A. Bjerkestrand, Elin T. Brødholt, and Carina V.S. Knudsen, the Governor of Svalbard, Svalbard Museum and NIKU.

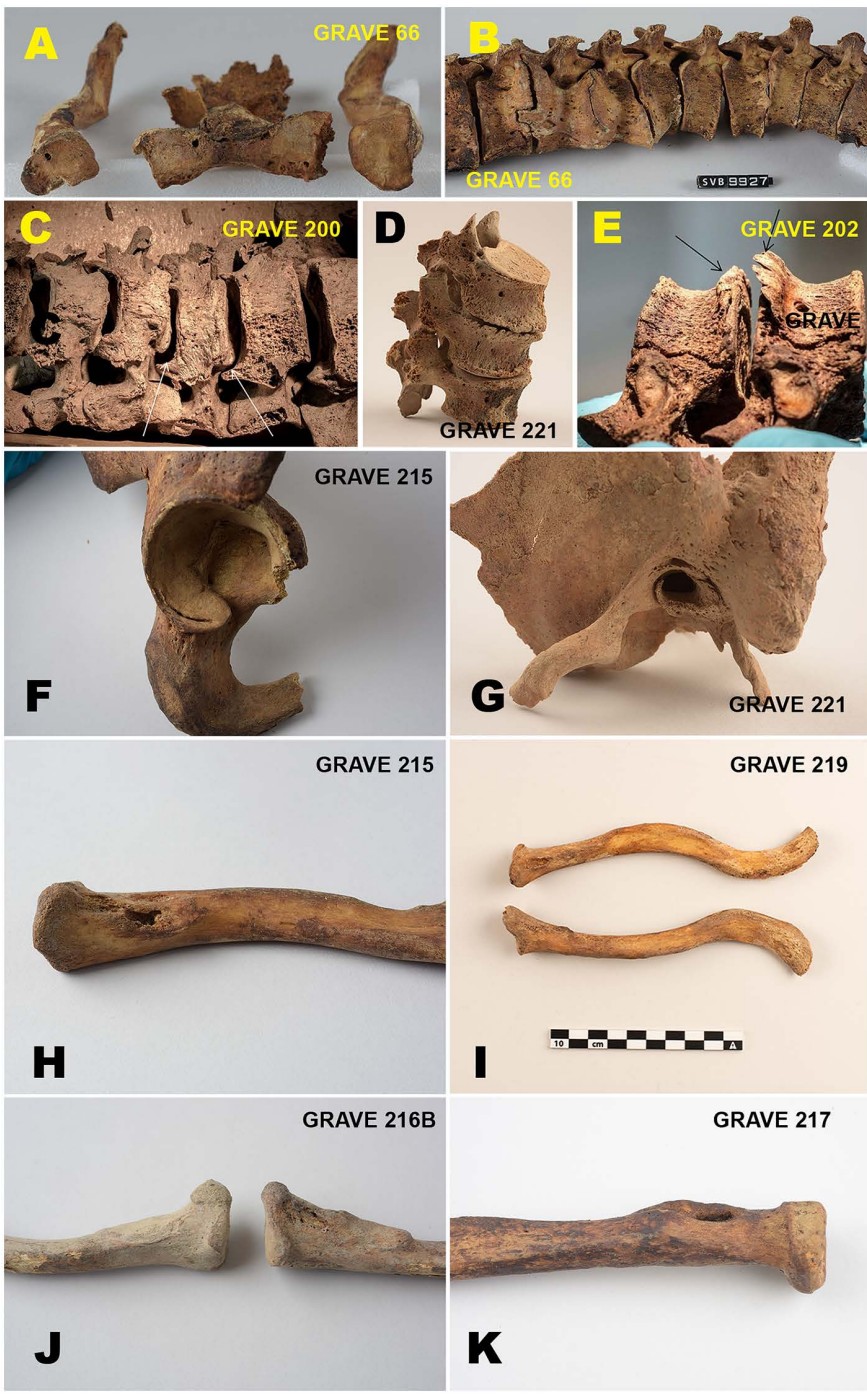

**Fig 14. Degenerative joint disease and activity-related skeletal stress in the Likneset assemblage.** Selected examples of degenerative and activity-related skeletal changes affecting both the axial and appendicular skeleton. The figure illustrates vertebral lipping, facet joint degeneration and disc-related changes in the spine (A–E), enthesopathic alterations at major muscle and ligament attachment sites (F), degenerative changes of the hip, long bones, and lower limb joints (H–K), and changes consistent with external auditory exostoses (G). The examples represent multi-regional involvement within individuals and reflect sustained mechanical loading and repetitive physical strain consistent with intensive maritime labour. Figure by Lise Loktu, NIKU. Photos by Lise Loktu, Mikael A. Bjerkestrand, Elin T. Brødholt, and Carina V.S. Knudsen, the Governor of Svalbard, Svalbard Museum and NIKU.

## Discussion

### RQ1: Climate, taphonomy, and the transformation of permafrost preservation

Preservation at Likneset is governed primarily by spatial variation in geomorphological stability within the burial site, with a clear and consistent contrast between the erosion-affected area (Field area A) and the more stable central area (Field area B). Graves in the erosion zone are typically characterised by damage associated with active geomorphological processes, including ground cracking, downslope sediment movement, and collapse of burial structures, whereas graves from the central area exhibit substantially better overall preservation.

**Preservation patterns.** Phase I burials (1985–1990) from Field area A (n = 14) show the greatest internal variability in preservation. Burial structures and coffins are frequently disturbed, while skeletal completeness remains high, although surface preservation is often reduced due to erosion-related processes. Textile preservation is highly variable but, where present, often good to exceptionally well preserved, indicating that organic materials were still largely intact at the time of excavation despite substantial structural disturbance.

The Phase II burials (2016), also from Field area A (n = 3), form a more constrained group characterised by intermediate burial and coffin preservation, likely reflecting slightly reduced cumulative exposure to sediment movement. In contrast to Phase I – even in immediately adjacent graves (notably 216A, 216B, 217, and 218) – textiles are largely absent, with only very small fragments preserved in Grave 201.

This contrast indicates progressive degradation of textiles under broadly comparable microenvironmental conditions. This likely reflects heightened sensitivity to prolonged exposure to moisture, oxygen, and repeated freeze–thaw cycles following earlier coffin collapse or sediment disturbance.

The Phase III burials (2019) from Field area B are characterised by consistently high preservation across all assessed variables. Burial structures and coffins are largely intact, skeletal preservation is uniformly good, and textile remains are still present, although generally less robust than those recovered during the 1980s. The resulting low variability in preservation scores reflects the relative geomorphological stability of this central part of the burial site.

Taken together, these patterns demonstrate that preservation at Likneset reflects both strong spatial contrasts within the burial site and temporal change within the erosion-prone zone. While burial structures in Phase II remain broadly comparable to those documented in the 1980s, the pronounced decline in textile preservation highlights the vulnerability of some organic materials to climate-driven geomorphological processes. In contrast, skeletal material reflects the greater resilience of mineralised tissues under unstable burial conditions.

**Likneset in a broader Arctic perspective.** The preservation trajectories documented at Likneset closely parallel findings from Greenland and other Arctic regions, where climate-driven permafrost thaw accelerates the degradation and loss of organic archaeological materials [2–6]. Across Arctic burial sites, middens, and settlement deposits, organic materials such as textiles, wood, and leather respond rapidly to increases in active-layer depth, moisture availability, and oxygen exposure [11–15]. Experimental and modelling-based studies further demonstrate strong temperature sensitivity in these processes, particularly under repeated freeze–thaw and wet–dry cycles [13,66–68].

Within this broader Arctic context, Likneset provides rare time-depth evidence for climate-driven preservation change at the scale of an individual burial site. Comparison of burials excavated over several decades documents environmental change directly within the archaeological record, complementing instrumental climate data and permafrost studies from Svalbard [16–24]. The pronounced decline in textile preservation in the erosion-prone zone between the 1980s and 2016 mirrors patterns reported elsewhere in the Arctic and supports model-based projections of selective organic loss under warming permafrost conditions.

Likneset thus functions both as a cultural archive and as an empirical indicator of climate-driven transformation in permafrost landscapes, illustrating how preservation outcomes can vary markedly over short distances and timescales. The site highlights the value of archaeological datasets for documenting environmental change and for integrating cultural heritage observations into Arctic climate research.

**Implications for heritage management and future research.**  Cultural heritage management on Svalbard, and in Norway more broadly, has long been guided by the principle that in situ preservation represents the preferred strategy [34,106]. Since 2013, management on Svalbard has been formally structured around a policy of managed decay, whereby a limited selection of particularly valued heritage assets, primarily standing buildings, are prioritised for maintenance, while most sites are allowed to deteriorate naturally with minimal intervention [34]. In practice, many of these sites are also only sparsely documented.

Under conditions of rapid climate change, this approach entails an increasing risk that archaeological heritage will be lost without the knowledge it contains ever being recorded. As permafrost thaws and erosion accelerates, archaeological contexts that were previously stable are now undergoing rapid and irreversible transformation.

Until recently, archaeological heritage on Svalbard has received limited systematic attention, aside from renewed excavations in the latter half of the 2010s. At present, no long-term monitoring programmes explicitly include archaeological sites, and erosion threats are therefore only partially documented [33–37]. Recent cases of rapid degradation at key sites have in some instances required emergency interventions, such as the relocation of the Pomor hut at Rekvika (site ID 93146-1) to Svalbard Museum [107].

The results from Likneset demonstrate that established management principles are now under significant strain and call into question the long-term viability of in situ preservation and managed decay under warming permafrost conditions. As archaeological materials are no longer preserved in permafrost environments as they once were, there is an urgent need to integrate archaeological sites more fully into political and strategic planning frameworks. This integration should be guided by clearly defined knowledge priorities: which information must be documented and analysed before it is irretrievably lost?

Likneset further illustrates that climate-driven heritage loss is highly uneven, with preservation potential varying at the scale of metres depending on microtopography and sediment conditions. Material-specific taphonomic assessment can therefore help identify areas at elevated risk, even where skeletal remains appear comparatively stable. At the same time, the pace of degradation increasingly exceeds the capacity for documentation and protection, raising the likelihood of irreversible information loss.

Reanalysis of legacy excavation material provides critical time-depth evidence for preservation change and complements modern permafrost and erosion monitoring. Applied more broadly, this approach enables comparison of preservation trajectories across sites and highlights the value of archaeological datasets as empirical indicators of environmental change. Effective responses to climate-driven heritage loss will require long-term monitoring, closer integration of research and heritage management, and stronger recognition of archaeological data within climate adaptation and sustainability planning. Protecting archaeological heritage should therefore be understood not only as preservation of the past, but as a contribution to resilience in rapidly changing Arctic landscapes.

### RQ2: Health, labour, and living conditions in early modern Arctic whaling

**Demography, labour selection, and workforce composition.**  The demographic composition of the Likneset burial assemblage is highly uniform. All analysed individuals were biologically male, which parallels findings from neighbouring whaling burial sites, including Ytre Norskøya, Jensenvatnet, and Smeerenburg, and accords with historical ship lists documenting whaling crews as male [41,43,44].

The age-at-death distribution indicates a workforce dominated by young adults, with a substantial proportion dying between 20 and 25 years of age. This relatively narrow age profile contrasts with that of Ytre Norskøya, where both younger and considerably older individuals are represented. As noted by Georg Maat [43], burial assemblages reflect not only who died, but also who survived long enough to leave the Arctic alive. Differential vulnerability to disease, malnutrition, and physical stress across age groups likely shaped the demographic profile of those ultimately buried.

Stature estimates indicate a relatively tall and physically robust group compared to neighbouring whaling burial sites. Mean stature at Likneset (173.3 cm) exceeds values reported from Ytre Norskøya (n = 50; mean 166 cm) and Jensenvatnet (n = 22; mean 166.6 cm) [43,50,108,109]. Such differences may reflect variation in socioeconomic background, nutritional conditions, or the representation of different ethnogeographic groups [110, 111] within the whaling workforce and have previously been interpreted as indicating recruitment from specific regions or close biological relationships [50]. However, aDNA analyses of individuals excavated in 2016 and 2019 reveal no evidence of close kinship among the sampled individuals [82, 83].

**Widespread metabolic stress and scurvy.** Scurvy represents the most pervasive metabolic condition in the Likneset assemblage, with skeletal indicators consistent with vitamin C deficiency recorded in 18 of 19 individuals. Lesions are dominated by periarticular and subperiosteal changes affecting the long bones, most frequently the femora, tibiae, fibulae, and humeri, while cranial involvement is rare. This pattern accords with established paleopathological observations that adult scurvy is primarily expressed in postcranial elements, with cranial lesions being less consistently developed than in juveniles [99,100,112].

Historical evidence suggests that many individuals likely arrived in the Arctic already vitamin-depleted following winters with limited access to fresh produce [43,99]. Contemporary accounts describe substantial mortality from scurvy during Arctic voyages, with severe debilitation and death occurring within months of departure [43,70–72,113,114]. Unlike Indigenous Arctic populations [115], European whalers relied largely on preserved provisions [116, 117], as reflected in cargo lists dominated by salted meat, hardtack, dried fish, butter, cheese, and grains [42,71]. Antiscorbutic foods appear to have been limited and socially differentiated, reserved for officers [71].

The skeletal features observed, likely linked to scurvy, suggest variable durations of vitamin C deficiency. Localized lesions, such as dark discoloration of joint ends, porous cortical bone surfaces, vascular impressions on tibial shafts, and periosteal reactions, may reflect shorter or more acute episodes, whereas generalized skeletal involvement and osteopenia in more severe cases are consistent with prolonged or recurrent deficiency. Comparable lesion patterns reported from Ytre Norskøya, Smeerenburg [43,72,99], and Jensenvatnet [108,109] indicate that scurvy was endemic among early modern Arctic whalers and support its interpretation as a structural health risk inherent to long-voyage maritime industries [116, 117].

**Evidence of diet.** Although isotopic data from Likneset remain limited, preliminary stable isotope analyses (strontium, oxygen, carbon, and nitrogen) from three individuals excavated in 2016 (Graves 200–202; analyses of the 2019 graves are ongoing), together with comparative data from two individuals excavated at Smeerenburg in 2017 [55], indicate marked variation in childhood origin and early-life diet, alongside a more homogeneous dietary pattern in adulthood [84].

Strontium and oxygen isotope values suggest that one individual from Likneset (Grave 202) and one from Smeerenburg (ID 93812−28) likely grew up in regions compatible with the present-day Netherlands, whereas individuals 200 and 201 from Likneset and 93813−24 from Smeerenburg show signatures consistent with colder climates and higher marine protein intake during childhood, pointing to more northerly coastal regions of Scandinavia, particularly along the Norwegian west coast.

Considered alongside radiocarbon dates placing these individuals in the late seventeenth to early eighteenth centuries [77, 78], the isotopic patterns align with historically documented labour mobility and maritime exchange between the Netherlands and southwestern coastal Norway [73]. Variation in oxygen isotope values between teeth formed at different stages of childhood further suggests residential mobility or access to food and water from multiple geographic regions.

In contrast, carbon and nitrogen isotope values from adult bone collagen indicate a convergent adult diet dominated by terrestrial animal protein with only limited marine input, and no evidence for substantial reliance on local terrestrial resources from Svalbard, such as reindeer [84]. This pattern accords with comparative isotopic studies of Dutch whalers from Ytre Norskøya, which document predominantly terrestrial childhood diets followed by a dietary shift in adulthood linked to repeated seasonal engagement in Arctic whaling [118].

**Developmental stress and life-course trajectories.** Dietary data gain additional context when considered alongside indicators of developmental stress. The high prevalence of enamel hypoplasia (63%) indicates that many individuals experienced episodes of illness or nutritional stress during childhood and adolescence [96,101]. While enamel hypoplasia records episodic early-life stress, adult stature reflects cumulative growth conditions extending into adolescence. Although no clear association is observed between early-life stress and adult stature, the frequent co-occurrence of hypoplasias and relatively tall adult stature suggests that early growth disruption may, in some cases, have been followed by partial or complete catch-up growth [119].

Selective recruitment into physically demanding maritime labour may have favoured individuals who, despite early-life stress, achieved relatively robust adult stature through genetic predisposition or successful growth recovery prior to employment. Isotopic data likewise indicate variation in childhood origin and early-life diet, reflecting diverse environmental and socioeconomic conditions during upbringing, followed by a more homogeneous dietary pattern in adulthood. This convergence may reflect a life-course transition into shared provisioning systems within maritime labour [118], potentially following periods of coastal seafaring or other forms of employment prior to entry into the more physically demanding and environmentally harsh conditions of Arctic whaling. Survivorship bias may also influence the burial assemblage, as the individuals represented at Likneset were those who survived both childhood adversity and the demands of maritime labour long enough to die and be interred in the Arctic.

Given the limited sample size and the absence of detailed information about individual life histories and employment trajectories, these interpretations cannot be tested directly with the present material. Further comparative osteological, historical, and biomolecular research will be required to clarify the relationship between early-life stress, adult stature, recruitment practices, and survival within early modern maritime labour systems.

**Musculoskeletal stress, labour, and everyday practices.** Degenerative joint disease and activity-related skeletal changes are nearly ubiquitous at Likneset (c. 95%) and affect multiple regions of the skeleton, including the spine, shoulders, upper limbs, hips, and lower limbs. Changes typically associated with advanced age occur frequently in young adults, indicating sustained physical strain from an early stage of working life.

Enthesopathic alterations involving the inferomedial clavicle, associated with the costoclavicular ligament, occur in 26% of individuals. Similar features have previously been linked to repetitive upper-limb loading, such as paddling-related activities [120, 121], although such changes are not activity-specific [122]. More broadly, over half of the assemblage (c. 53%) exhibit complex, multi-regional degeneration involving the shoulder girdle and upper limbs, consistent with intensive and prolonged upper-body loading.

Dental wear patterns indicate habitual non-masticatory use of the teeth, while clay pipe smoking was widespread, in line with observations from neighbouring whaling burial sites [43,46,51,108,109]. Tobacco use was common among seafarers [123] and likely served recreational, social, and possibly analgesic functions within the physically demanding context of Arctic whaling [124].

**Cumulative pathology and lived experience.** Taken together, the osteological evidence documents a pattern of cumulative, multi-systemic physiological strain. Most individuals exhibit multiple co-occurring stress indicators, including metabolic disease, developmental stress markers, degenerative joint changes, musculoskeletal strain, dental pathology, and healed injuries.

Importantly, pathological burden does not increase linearly with age. Several young adults show skeletal involvement comparable to, or exceeding, that of middle adults, indicating that severe physiological stress could develop early in adulthood. This pattern reflects the combined effects of intense physical labour, nutritionally constrained provisioning, and repeated metabolic stress, rather than age alone.

Overall, the Likneset assemblage represents a relatively homogeneous group shaped by structured recruitment and shared labour conditions within early modern Arctic whaling. Despite diverse childhood backgrounds, adult diets and working lives appear to have converged within common maritime provisioning systems. The predominance of healed

injuries further suggests that mortality was more closely linked to cumulative physiological stress than to isolated traumatic events.

## Conclusion

Rapid Arctic warming is accelerating the degradation of permafrost-preserved archaeological sites, placing organic-rich whaling burials on Svalbard among the most vulnerable heritage contexts. This study integrates taphonomic and osteological analyses of the Likneset burial site to examine both climate-driven preservation change (RQ1) and the embodied health costs of labour in early modern Arctic whaling (RQ2).

Preservation at Likneset is shaped by both spatial variability and ongoing temporal change. Graves in geomorphologically unstable areas (A) show extensive disturbance and rapid loss of organic materials, whereas burials in more stable settings (B) retain well-preserved structures, skeletons, and textiles. Comparison of burials excavated decades apart within the same erosion-exposed area shows a clear decline in textile preservation despite similar burial conditions, highlighting the sensitivity of organic materials to climate-driven processes. In contrast, skeletal remains are more resilient, resulting in divergent preservation trajectories within the same site and providing rare time-depth evidence of climate-related degradation. These findings challenge the long-term viability of in situ preservation within Svalbard's cultural heritage management and highlight the importance of systematic monitoring and targeted documentation of vulnerable sites. Reanalysis of legacy excavation material provides a valuable approach for understanding local preservation processes over time.

Osteological evidence identifies a highly uniform burial population of predominantly young adult men, consistent with structured recruitment into Arctic whaling. Although individuals are relatively tall and robust, suggesting selective recruitment, the skeletal record documents pervasive physiological stress. The high prevalence of scurvy reflects chronic nutritional deficiency within a provisioning system poorly adapted to Arctic conditions. Isotopic data indicate diverse childhood origins followed by convergence in adult diet, consistent with a mobile, multinational workforce. Developmental stress markers, combined with adult stature, suggest partial catch-up growth prior to entry into Arctic labour, while widespread degenerative and activity-related skeletal changes, including among young adults, indicate intensive physical strain from an early stage of working life. Trauma is predominantly healed, suggesting that mortality was driven primarily by cumulative metabolic stress and prolonged physical strain rather than acute injury.

Overall, the Likneset assemblage reveals how early-life stress, nutritional deficiency, and sustained physical labour accumulated across the life course, providing a bioarchaeological record of the embodied health costs of early modern Arctic whaling under extreme environmental and occupational conditions.

Despite its limited sample size, the Likneset assemblage highlights the potential of integrated osteological, archaeological, and environmental analyses. Future research should prioritise comparative studies across burial sites in the Smeerenburgfjorden region to refine interpretations of recruitment, labour organisation, and social composition in early modern Arctic whaling. Continued monitoring of preservation conditions is also essential, as climate-driven degradation and coastal erosion are rapidly reducing the informational value of archaeological archives on Svalbard.

## Supporting information

**S1 Appendix A. Burial preservation and contextual assessment dataset.** Excel dataset containing detailed semi-quantitative scoring and contextual observations for all recorded graves and individuals from Likneset, including excavation phase, grave number, field area, burial structure integrity, coffin integrity, coffin fill, textile preservation, skeletal completeness, skeletal preservation, and associated taphonomic notes.
(XLSX)

**S2 Appendix B. Osteological and pathological assessment dataset.** Excel dataset containing individual-level osteological observations for the analyzed Likneset assemblage, including skeletal completeness and preservation, sex and age assessment, stature estimates, dental observations, pathological changes, activity-related skeletal indicators, and selected biomolecular and radiocarbon variables where available.
(XLSX)

**S1 Table. Repository and specimen numbers, excavation history, and osteological analysis overview for all individuals from the Likneset burial site (site ID 93705).** Word table listing all analyzed individuals from the Likneset burial site, including excavation phase, year, grave number, specimen or museum number, field area, excavation method, and summary information on osteological analysis and curation history.
(DOCX)

## Acknowledgments

We gratefully thank our colleagues at the Svalbard Museum for facilitating the analyses and for their valuable academic collaboration. Special thanks are extended to Mikael A. Bjerkestrand for assistance with laboratory analyses and for his in-depth knowledge of the excavation material. We also thank the Governor of Svalbard for granting access to excavation data and images, and for their professional support. Finally, we thank Alma Thuestad at the Norwegian Institute for Cultural Heritage Research (NIKU) for her helpful comments and careful proofreading.

## Author contributions

**Conceptualization:** Lise Loktu.

**Data curation:** Lise Loktu.

**Formal analysis:** Lise Loktu, Elin Therese Brødholt.

**Funding acquisition:** Lise Loktu.

**Investigation:** Lise Loktu, Elin Therese Brødholt.

**Methodology:** Lise Loktu, Elin Therese Brødholt.

**Project administration:** Lise Loktu.

**Resources:** Lise Loktu.

**Visualization:** Lise Loktu.

**Writing – original draft:** Lise Loktu.

**Writing – review & editing:** Lise Loktu, Elin Therese Brødholt.

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
