## [Decision Letter · Decision Letter 0]

28 Oct 2025

Dear Dr. Loktu,

Thank you for submitting your manuscript to PLOS ONE. After careful consideration, we feel that it has merit but does not fully meet PLOS ONE’s publication criteria as it currently stands. Therefore, we invite you to submit a revised version of the manuscript that addresses the points raised during the review process.

Both reviewers agree that the study addresses a highly relevant and timely topic — the impact of climate change on Arctic archaeological heritage — and that the Likneset site represents a unique and valuable case study.

The current version reads largely as a descriptive site report, with limited integration between the stated research aims, the methods applied, and the results presented. Both reviewers highlight the need for clearer methodological detail—particularly regarding sampling strategy, data comparability across excavation phases, and taphonomic recording—and for a more explicit, evidence-based link between observed degradation and climate-related processes. They also emphasize the need for more quantitative data, a clearer separation between results and discussion, and a stronger articulation of the study’s implications for heritage management. What is really lacking is the data (published and original) integration with a proper separation between results and discussion.

We look forward to receiving your revised manuscript.

Kind regards,

Alessia Nava, Ph.D.

Academic Editor

PLOS ONE

Journal Requirements:

2. In your manuscript, please provide additional information regarding the specimens used in your study. Ensure that you have reported human remain specimen numbers and complete repository information, including museum name and geographic location.

For more information on PLOS One's requirements for paleontology and archeology research, see https://journals.plos.org/plosone/s/submission-guidelines#loc-paleontology-and-archaeology-research....

3. Please note that your Data Availability Statement is currently can’t access a direct link to each database. If your manuscript is accepted for publication, you will be asked to provide these details on a very short timeline. We therefore suggest that you provide this information now, though we will not hold up the peer review process if you are unable.

Additional Editor Comments:

I suggest, as hjghlighted by reviewer 2, to ass the scale bars in the images where lacking.

Figure 1: Low resolution; not clear the legends, add explanation of the legend in the caption, each coloured dot represent a site or a single burial?

From the manuscript, as also highlighted by reviewer 2, it seems that the author performed a diachronic analysis of the burials, with the meaning of difference in time between archaeological excavations. I suggest to not refers as “diachronic” in this case, it is measleading, unless the diffrent excavation campaigns involved different time-period burials.

Figure 4: the panel letter are not clearly visible, please enlarge the font size.

Reviewers' comments:

Reviewer's Responses to Questions

**Comments to the Author**

1. Is the manuscript technically sound, and do the data support the conclusions?

Reviewer #1: Yes

Reviewer #2: No

2. Has the statistical analysis been performed appropriately and rigorously?

Reviewer #1: N/A

Reviewer #2: No

3. Have the authors made all data underlying the findings in their manuscript fully available?

Reviewer #1: Yes

Reviewer #2: No

4. Is the manuscript presented in an intelligible fashion and written in standard English?

Reviewer #1: Yes

Reviewer #2: Yes

Reviewer #1: The study “Skeletons in the Permafrost: Exploring Climate-Threatened Burial Preservation and Occupational Health in an Early Modern Whalers’ Burial site at Likneset, Svalbard” examines skeletal indicators of health and occupational stress in individuals from Likneset, a burial site in Svalbard, Norway, as well as the diminished preservation of the site’s organic remains with progressive climate change. In an analysis of 19 individuals, the results of the osteological analyses are consistent with the repetitive biomechanical strain associated with maritime labor and lifelong physiological stress. Additionally, multiple distinct site zones allow the authors to evaluate the differential impacts of climate change across the site and demonstrate the need for interdisciplinary preservation strategies to safeguard the material records of the lives and experiences of early modern Arctic whalers. I enjoyed reading this manuscript and have a few requests for clarification and suggestions for revision, which are provided below.

Page 4, Line 128: I see on the title page (page 1) that NIKU stands for Norwegian Institute for Cultural Heritage Research, but it may be helpful to readers unfamiliar with the Institute if the acronym is spelled out for the first time in the text as well.

Page 8, Lines 267-268: Generally, cranial suture closure and dental wear are considered less precise/reliable methods for estimating age at death, especially given the sources of tooth wear discussed later in the paper (i.e., habitual pipe use). Were ages estimated using the pubic symphysis given precedence, when possible?

Page 8, Line 283: This sentence begins “Taphonomic scoring consistently applied…” and is followed by a list of relevant variables of interest. Should this sentence read “Taphonomic scoring was consistently applied…”?

Page 10, Lines 323-324: “Compared to average statures at Ytre Norskøya and Jensenvatnet, which range 166 (n=50) and 166.6 cm (n=22)....” The phrasing in this sentence is somewhat confusing, since the word “range” was used to refer to a distribution of heights in a preceding sentence. Perhaps one of the following options would work: “...which were 166 cm (n=50) and 166.6 cm (n=22)...” or “...which were estimated to be 166 cm (n=50) and 166.6 cm (n=22)...”.

Page 10, Line 332: “While these supports a multinational interpretation…” There appears to be a word missing here. Could the authors rephrase? For example: “While these findings support…” or “While these results support…”

Pages 10-11: The last paragraph on page 10 (Lines 337-340) and the first paragraph on page 11 (Lines 364-366) contain nearly identical descriptions of the symptoms associated with scurvy. I recommend the second iteration be removed, given that the focus of the paragraph is historical accounts of scurvy on long sea voyages.

Page 12, Lines 407-408: I think the hypothesis about the discrepancy between enamel hypoplasia prevalence and relatively tall adult stature is interesting and worth exploring. Particularly, I wonder what sorts of training/occupations were available to adolescent individuals (i.e., who would eventually become the young or middle-aged adults hired for whaling crews), as these are the contexts that would have given them a chance to experience catch-up growth in adolescence.

Page 13, Lines 442-443: “...a condition previously associated with individuals engaged in cold-climate maritime occupations.” Could the authors provide one or more citations for this statement?

Page 14, Line 475: This section highlights the importance of “maintaining stable conservation environments and adhering to standardized post-recovery protocols”. Could the authors provide references to other studies of post-excavation degradation or published protocols?

Page 15, Line 509: “...historically associated with military dress and social hierarchy in the 17th century.” Reference for this statement?

Reviewer #2: This is an interesting and relevant report on a whaler-context burial ground in the Arctic. The connection drawn between climate change and the urgency of preserving vulnerable archaeological sites is both timely and valid. Overall, the paper is clearly written, and its objectives are coherent and well-framed.

My main concern lies in the (absence of) depth and analytical rigor of the study. Much of the paper is presented descriptively, relying on qualitative summaries rather than systematic analysis. The manuscript reads as a well-contextualized osteological site report (but with a missing data appendix) rather than an ‘interdisciplinary analysis’ as it is stated in the introduction. The authors refer to having performed a ‘diachronic’ analysis, which sounds like the analysis is done for burials from different time periods, whereas in fact, here with diachronic, it is meant that graves excavated in different campaigns are compared. Which is also of interest, but this comparison is rather skewed: 16 graves in one group, 3 in the other... Furthermore, both in the abstract and introduction it is stated it concerns an interdisciplinary study, but the interdisciplinary data (e.g. istoptic and DNA) are from other studies and only referred to here, and not integrated. Climate change is only referred to in general terms, without clear references. The conclusions on the demographics and health conditions of the whalers are consistent with expectations for early modern Arctic whalers but do not extend beyond what is already established in the literature which, by it self is ok, but the lack of clarity in the results makes that some work is needed to make this paper ready for publication. The statement that it is likely that many individuals arrived in the Arctic already vitamin-depleted following winters of poor access to fresh produce is of interest, but no clear analytical base here. There are ways to examine this, but none of that is referred to. I also recommend taking a look at the work of Swanston on the Franklin expedition (I know this is ofcourse from a later phase of the waling expedition, but here they are also discussing ways of interring health before and during the expedition).

I do like how this report gives extra strength to what we know of the antiscorbutic measurements that were taken in the later stages of the whaling expeditions, and here it's very clear how osteological evidence can supplement historical sources. Whereas for example the work of Mays, Maat & The Boer 2012 had to conclude that the material they examined must have been from the period where antiscorbutic measurements were already (better) in place, the study presented here is clearly from before. I would reference this and the work of Mays 2012 clearer in paragraph 382-387.

With some reworking of the more detailed reporting of the osteological evidence, and a similar more systematic approach for the taphonomy description, and general clarity and addition of appendix of obtained measurements/scorings of features, this paper could be made ready for publication

Some more specific notes:

Prevalence of scurvy is reported as extremely high, the description focuses on skeletal manifestations associated with ‘advanced deficiency’, yet no systematic scoring of lesions is applied. There is no mention made of periodontal involvement, which could also be an early signs especially when other extreme features such as hemorrhaging observed. It remains unclear whether some features were not observed, not preserved, or simply not analyzed. No diagnostic criteria or lesion scoring methods are provided, making it difficult to assess the robustness of the pathological interpretations.

Similarly, the discussion of enamel hypoplasia as evidence of early-life stress followed by compensatory growth (as inferred from stature) is an intriguing comment but underdeveloped. The statement would benefit from grounding in relevant bioarchaeological or developmental literature, as well as a more robust description of the enamel defects themselves (type, distribution, affected teeth, frequency). As it stands, these observations remain anecdotal rather than analytical.

While the authors refer to aDNA and isotopic data, these appear largely drawn from previous publications and are not integrated quantitatively into this study – even though they are referred to ‘in the context of ‘interdisciplinary’ nature of the study

It’s not clear to me if the published isotopic data from Kootker et al (56) and the aDNA data (55) are from the same site context, or from nearby. If they are from the same site, then why is the data not integrated with the results here? That could be of high interest, especially as there are hints made towards pre-whaling poorer living conditions, and possibly different geographic origins of the whalers. Also, if not from the same site, are there during the osteological analysis efforts made for sampling to allow future isotopic or DNA analysis or other advanced analytical techniques? As one of the core objectives of the paper seems to be to highlight the issue of degradation, are there preventative measures taken so that future analytical techniques can still take place? I can read hints towards this in the paper but it's not clearly outlined

Line 62: - add additional relevance references (including work by G. Maat - it appears later, but already relevant here). Also, check spelling of Smeerenburg, as in figure its written as Smeereburg

Line 337-340 and line 364-366 are direct repeats

Line 392- ‘most commonly occurring between age 2 and 15 – why not be more precise here on the interpretation?

Line 405 ‘relatively tall adult stature’ – this section is underdeveloped: as 63% have enamel hypoplasia but it is not cross-linked with stature, even though interpretations on the life-course are based on the combined data

From line 264, can I conclude that all skeletal remains included in the study have been re-analysed or only the ones from the older campaigns? Are the more recently excavated remains not re-analysed/ assessed by the same authors? It’s not completely clear if the osteological data for the 19 individuals has all been re-analysed by the authors of this paper, or if it’s a combination of past and renewed efforts. Please clarify.

Line 340 says that 95% of individuals have signs of advanced ‘deficiency’, while the caption of Fig 4 says 78 per cent. There is no systematic scoring presented of lesions provied, only overarching statements and a few images. “Ortner Criteria” or the synthesis work from Snoddy et al 2018 are mentioned but not systematically adhered to. Since the bioarchaeological literature on adult scurvy is relatively small, a much more detailed description of the pathological lesions and their exact locations and manifestations would make this manuscript much stronger, and allow it to go beyond a mere site report. Instead, many images are presented of pipenotches, which, although a clear ‘teeth-as-tool’ marker, much clearer in assessment and in need of less visual backing. I would rather see more in-depth description and visuals of the porotic manifestations. It almost seems as if an appendix with systematic scoring data of osteological lesions and other osteological data is omitted.

The same lack of depth goes goes for the taphonomic assessment.

In line 283, the authors speak of ‘taphonomic scoring’, but without clearly stating what is scored: it says including variables ‘such as’.

Line 466 ‘significantly’ better, significantly means statistically visible, while no statistical data is presented. Replace with ‘notably’.

488 ‘ preliminary analysis of the fibers' can you specify what kind of analysis? It seems there is being referred to a report currently underway.

503: animal fibers generally better preserved than plant based fibers: again no real quantification but a mere note or general observation.

549: again ‘significant better preservation’ without any clear backing data on this.

553: shallower depth : are there any measurements available?

51: climatic conditions… were colder.. – there is no data presented nor any references.

The whole section from 561 to 565 lacks references.

Paragraph 658-573: same problem, very anecdotally written without referencing any sources or detailed information

Figures: Scales are absent from many images. The osteological images have no scales at all, and figure 3 as a site overview should have a clear scale to assess the size of the site.

Figure 1: labels are in Norwegian on the image – change to English

.

Reviewer #1: No

Reviewer #2: No

---

## [Author Response · Author response to Decision Letter 1]

12 Jan 2026

Response to Reviewers – [PONE-D-25-37512]

Reviewer / Editor comment Response Location in revised manuscript

Academic Editor: Manuscript reads largely as a descriptive site report with limited integration between aims, methods, and results. The manuscript has been substantially revised to strengthen analytical integration. Aims, methods, and results are now explicitly aligned, with Results structured according to the two research questions (RQ1 and RQ2), and Discussion synthesising findings without introducing new data. Introduction; Methods (Analytical framework); Results (section structure); Discussion

Academic Editor: Limited integration between research aims, methods, and results. This integration has been clarified throughout. The analytical framework is now explicitly designed to address the stated research questions and enable comparison across excavation phases and burial microenvironments. Introduction; Methods; Results

Academic Editor: Need clearer methodological detail regarding sampling strategy, data comparability across excavation phases, and taphonomic recording. We now explicitly state that all 19 individuals included in the osteological analyses were reassessed by the same analyst using identical protocols, including material excavated in the 1980s. A detailed semi-quantitative preservation and taphonomic scoring system is presented, with variables defined in Table 1 and individual-level data provided in the Supplementary Data. Methods: Osteological methods; Analytical framework and preservation assessment; Table 1; Supplementary Data (Appendices A–B)

Academic Editor: Need a more explicit, evidence-based link between observed degradation and climate-related processes. The manuscript now clarifies that the study documents empirical preservation change rather than direct climatic causation. Comparison between Phase I (1985–1990) and Phase II (2016) burials is explicitly restricted to graves located in close spatial proximity within the same erosion-exposed sector (Field area A), documenting continued in situ degradation under broadly comparable microenvironmental conditions. This point is now clearly stated in the Abstract, Results, and Conclusion. Abstract; Results (RQ1); Conclusion

Academic Editor: Need more quantitative data. Quantitative and semi-quantitative data are now presented through tables and supplementary appendices, including prevalence values and preservation scores. Results; Tables; Supplementary Data

Academic Editor: Clearer separation between Results and Discussion required. Results and Discussion have been fully separated. Observations and descriptive data are confined to the Results section, while interpretation and synthesis are restricted to the Discussion. Results; Discussion

Academic Editor: Stronger articulation of implications for heritage management. The Discussion and Conclusion now explicitly address implications for heritage management under warming permafrost conditions, including the limitations of in situ preservation and the value of legacy excavation data for monitoring degradation. Discussion; Conclusion

Academic Editor: Figures require improvements (scale bars, resolution, legends, terminology). Scale bars have been added where possible; figure resolution improved; Norwegian labels replaced with English; figure legends clarified; panel labels enlarged. The term “diachronic” has been removed to avoid temporal ambiguity. Figures 1–10 and captions

Reviewer 1: Clarify acronym NIKU at first mention. The acronym is now spelled out at first occurrence in the main text. Introduction

Reviewer 1: Clarify age-at-death estimation methods. The Methods section now explicitly states that pubic symphysis morphology was given precedence where observable, with other indicators used complementarily. Methods: Osteological methods

Reviewer 1: Typographical correction (“Taphonomic scoring consistently applied…”). The original text has been removed and replaced with a fully revised section Methods

Reviewer 1: Clarify stature phrasing using “range”. The sentence has been rephrased to avoid ambiguity, using explicit mean values. Results: Stature

Reviewer 1: Missing word (“While these supports…”). Corrected to “While these findings support…”. Results / Discussion

Reviewer 1: Repetition of scurvy symptom descriptions. This section has been substantially rewritten to improve clarity, analytical depth, and consistency with the revised methodological framework. Results / Discussion

Reviewer 1: Expand and reference catch-up growth hypothesis. This section has been expanded, framed more cautiously, and grounded in relevant bioarchaeological and historical literature on adolescent labour, mobility, and growth. Discussion: Developmental stress

Reviewer 2: Manuscript lacks analytical depth and reads as a site report. The manuscript has been reframed as an analytical study. Semi-quantitative data are now presented in tables and appendices, and Results and Discussion are clearly separated. Methods; Results; Discussion

Reviewer 2: Use of the term “diachronic” is misleading. The term has been removed and replaced with explicit reference to comparisons between excavation phases. Abstract; Introduction; Discussion

Reviewer 2: Skewed sample size between excavation phases. The revised manuscript clarifies that the material derives from three excavation campaigns and that the value of the dataset lies in well-documented preservation differences across time and microenvironmental settings, rather than balanced sample sizes. Methods; Results (RQ1)

Reviewer 2: Extremely high scurvy prevalence without clear criteria or scoring. Diagnostic criteria are now explicitly stated with reference to established frameworks. Lesion presence, distribution, and manifestation are described systematically, with individual-level data provided in the Supplementary Data. Methods; Results (Metabolic disease); Supplementary Data

Reviewer 2: Enamel hypoplasia observations underdeveloped. Descriptions of defect types, affected teeth, and frequencies have been expanded and explicitly linked to life-course interpretation and stature data. Results; Discussion

Reviewer 2: Isotopic and aDNA data referred to but not integrated. The manuscript now clearly states that isotopic and aDNA data derive from the same site context, sampled during excavation, and are used here for contextual interpretation. These datasets will be presented in dedicated publications, and their future analytical potential is discussed. Methods: Biomolecular analyses; Discussion

Reviewer 2: Clarify whether all skeletal remains were re-analysed. It is now explicitly stated that all individuals included in the study, including those excavated in the 1980s, were reassessed by the same analyst using the same protocols. Methods: Osteological methods

Reviewer 2: Use of “significant” without statistical testing. All such terminology has been revised (e.g. replaced with “notable”) to avoid implying statistical inference. Results; Discussion

Journal requirements: Specimen numbers, repository information, permits, and data availability. Specimen numbers, repository information, permit details, and an updated Data Availability Statement are now provided. Methods; Data Availability Statement

---

## [Decision Letter · Decision Letter 1]

23 Feb 2026

Dear Dr. Loktu,

Thank you for submitting your manuscript to PLOS ONE. After careful consideration, we feel that it has merit but does not fully meet PLOS ONE’s publication criteria as it currently stands. Therefore, we invite you to submit a revised version of the manuscript that addresses the points raised during the review process.

We look forward to receiving your revised manuscript.

Kind regards,

Alessia Nava, Ph.D.

Academic Editor

PLOS One

Journal Requirements:

Additional Editor Comments:

The authors did very well in addressing all the reviewers' comments.

Reviewer 2 reccommend a minor revision on the discussion, and I agree with the their comment.

Reviewers' comments:

Reviewer's Responses to Questions

**Comments to the Author**

Reviewer #1: All comments have been addressed

Reviewer #2: (No Response)

2. Is the manuscript technically sound, and do the data support the conclusions?

Reviewer #1: Yes

Reviewer #2: Yes

3. Has the statistical analysis been performed appropriately and rigorously?

Reviewer #1: N/A

Reviewer #2: N/A

4. Have the authors made all data underlying the findings in their manuscript fully available?

Reviewer #1: Yes

Reviewer #2: Yes

5. Is the manuscript presented in an intelligible fashion and written in standard English?

Reviewer #1: Yes

Reviewer #2: Yes

Reviewer #1: (No Response)

Reviewer #2: The manuscript is much improved compared to the previous version. Great to see a lot more detailled osteological assessment and also the manuscript now has a much clearer focus with the dual goal as described in the abstract, much better reflecting the content of the manuscript and without overpromising what the goal is of the work. I feel most of the comments are well addressed, and also there is a much better section now on the implications of heritage management.

I think the main point I would suggest the authors to look at another time is a point in the discussion, where I feel there is some conflict in interpretations.

It concerns the point where the authors propose a 'catch-up growth hypothesis' to explain the presence of enamel hypoplasia but with tall statures. I think the discussion here needs some further reconsideration in light of the broader evidence the authors present themselves in this study.

Currently, it has the suggestion that "entry into maritime labour or related forms of adolescent employment may have provided improved nutritional intake and physical conditioning" (lines 888-889). But to me, this seems quite inconsistent with the substantial body of evidence the authors present elsewhere throughout the manuscript, documenting the severe nutritional and physiological stress experienced by Arctic whalers... Specifically:

- Scurvy affected 18 of 19 individuals, indicating chronic vitamin C deficiency

- Dietary provisions consisted primarily of preserved foods lacking fresh produce

- Antiscorbutic foods were socially restricted and unavailable to ordinary crew members

- Contemporary accounts document whalers arriving already vitamin-depleted and experiencing severe debilitation within months of departure

- Nearly all individuals (c. 95%) showed degenerative joint disease, with young adults exhibiting skeletal pathology comparable to much older individuals

These findings demonstrate that Arctic whaling represented a regime of cumulative physiological stress rather than improved living conditions.. I would think?

I do appreciate that the authors say that the 'improved living circumstances' are an hypothesis and that they cannot test it, but aside from the cautionary statement, maybe the authors could consider alternative explanations for the co-occurrence of enamel hypoplasia and relatively tall adult stature, I name a few:

Selection effects: Recruitment into physically demanding maritime labour may have favored individuals who, despite experiencing childhood stress, possessed sufficient height and physical robustness? And those could be determined either through genetic factors or successful growth recovery prior to employment rather than as a consequence of it.

Survival bias: The burial assemblage may preferentially represent individuals who were physically resilient enough to both survive childhood adversity and endure whaling labour long enough to die and be buried in the Arctic, rather than succumbing earlier

Just a few thoughts.

Other than that - I'd say the authors did a great job with the revisions

.

Reviewer #1: No

Reviewer #2: **Yes:** Simone A.M. LemmersSimone A.M. LemmersSimone A.M. LemmersSimone A.M. Lemmers

---

## [Author Response · Author response to Decision Letter 2]

17 Mar 2026

Response to Reviewer #2

Reviewer comment:

The reviewer noted a potential inconsistency in the discussion of catch-up growth, suggesting that the interpretation that entry into maritime Labouré may have improved living conditions appears inconsistent with the broader evidence presented in the manuscript showing substantial physiological stress among Arctic whalers. The reviewer suggested considering alternative explanations such as selective recruitment and survivorship bias.

Response:

We thank the reviewer for this thoughtful and helpful comment. In response, we have revised the discussion to clarify that any potential growth recovery would most likely have occurred prior to participation in Arctic whaling, which typically began in late adolescence or early adulthood. The previous formulation suggesting that entry into maritime labour itself may have improved living conditions has therefore been removed.

The revised text now situates this interpretation within a broader life-course perspective, noting that some individuals may have engaged in coastal or regional maritime activities, or other forms of labour, before entering the more physically demanding and environmentally harsh conditions associated with Arctic whaling.

In addition, we have incorporated the reviewer’s suggestions by explicitly discussing the potential role of selective recruitment into physically demanding maritime labour, as well as the influence of survivorship bias in shaping the burial assemblage. These revisions clarify that catch-up growth represents only one possible explanation and that alternative interpretations must also be considered in light of the limitations of the available data (see revised section “Developmental Stress and Life-Course Trajectories”).

---

## [Editor Report · Decision Letter 2]

26 Mar 2026

Skeletons in the Permafrost: Exploring Climate-Driven Heritage Loss and Occupational Health at the Early Modern Whaling Burial Site of Likneset, Svalbard

PONE-D-25-37512R2

Dear Dr. Loktu,

We’re pleased to inform you that your manuscript has been judged scientifically suitable for publication and will be formally accepted for publication once it meets all outstanding technical requirements.

Kind regards,

Alessia Nava, Ph.D.

Academic Editor

PLOS One

Additional Editor Comments (optional):

The authors have made a great effort to address the reviewers’ comments and revise the manuscript accordingly. I believe the paper has improved considerably, and I am pleased to accept it for publication.
---

## [Editor Report · Acceptance letter]

PONE-D-25-37512R2

PLOS One

Dear Dr. Loktu,

I'm pleased to inform you that your manuscript has been deemed suitable for publication in PLOS One. Congratulations! Your manuscript is now being handed over to our production team.

Kind regards,

on behalf of

Dr. Alessia Nava

Academic Editor

PLOS One